# Costs and health benefits of the rural energy transition to carbon neutrality in China

Teng Ma[1], Silu Zhang[1], Yilong Xiao[1], Xiaorui Liu[1], Minghao Wang[2], Kai Wu[1], Guofeng Shen [3,4], Chen Huang[1,5], Yan Ru Fang[1] ✉ & Yang Xie [2,6] ✉

The rural energy transition is critical in China's efforts to achieve carbon neutrality and improve air quality. However, the costs and health benefits associated with the transition to carbon neutrality remain unclear. Here we explore the cost-effective transition pathways and air quality-related health impacts using an integrated energy-air quality-health modeling framework. We find that decarbonizing rural cooking and heating would triple contemporary energy consumption from 2014 to 2060, considerably reducing energy poverty nationwide. By 2060, electric cooking ranges and air-to-air heat pumps should be widely integrated, costing an additional 13 billion USD nationally in transformation costs, with ~40% concentrated in Shandong, Heilongjiang, Shanxi and Hebei provinces. Rural residential decarbonization would remarkably improve air quality in northern China, yielding substantial health co-benefits. Notably, monetized health benefits in most provinces are projected to offset transformation costs, except for certain relatively lower-development southwestern provinces, implying more financial support for rural residents in these areas will be needed.

China is home to the second largest rural people worldwide, with approximately 550 million people consuming about 163 million tons of oil equivalent (Mtoe) of commercial energy in the residential sector in 2019[1,2]. Additionally, rural residents still extensively use traditional biomass and coal as the dominant energy carriers for livelihood-sustaining services such as cooking and heating[3,4]. Consequently, rural residential carbon dioxide ($CO_2$) emissions have increased annually by 4.8% from 2000 to 2017, accounting for 42% of China's total residential energy-related $CO_2$ emissions in 2017[5], while energy use of rural cooking and heating (RCH) accounted for 72% of China's total residential cooking and heating energy use in 2014[6]. Meanwhile, rural residential energy use contributed disproportionally high levels of air pollutants emissions, leading to over 3.9 μg/m³ ambient $PM_{2.5}$ in 2014[6] that harms human health[6–10]. Therefore, switching rural residential energy toward contemporary energy is pivotal to simultaneously attain multi Sustainable Development Goals (SDGs) of clean energy

(SDG 7), rural revitalization driven by energy poverty improvement (SDG 1) and universal contemporary energy access (SDG 10), carbon neutrality (SDG 13), better air quality related good health (SDG 3). Historically, rural China has transitioned from solid fuels to contemporary energy like electricity and gases, especially in well-developed regions[11]. Owing to a notable nationwide decrease of 50% in rural traditional biomass use like wood and crop residue[3], conventional air pollutant emissions have declined drastically. For instance, in rural North China, the scattered coal substitution policy has notably increased natural gas and electricity for space heating and cooking[12,13].

Existing retrospective studies found that the historical rural residential energy transition toward contemporary energy has brought multiple co-benefits in improving air quality and human health. For instance, solid fuels substitution and stove upgrading during 1980–2021 contributed to the national and regional reductions in $PM_{2.5}$ emissions, ambient and indoor $PM_{2.5}$ concentrations, population

[1]College of Environmental Sciences and Engineering, Peking University, 100871 Beijing, China. [2]School of Economics and Management, Beihang University, 100191 Beijing, China. [3]College of Urban and Environmental Sciences, Laboratory for Earth Surface Processes, Peking University, 100871 Beijing, China. [4]Institute of Carbon Neutrality, Peking University, 100871 Beijing, China. [5]International Institute for Applied Systems Analysis (IIASA), Laxenburg, Austria. [6]Laboratory for Low-carbon Intelligent Governance, Beihang University, 100191 Beijing, China. ✉e-mail: fyr000@126.com; xieyangdaisy@buaa.edu.cn

exposure, and premature deaths attributable to the rural residential sector[14,15]. When comparing human health benefits with energy transition costs, some energy policies, such as the residential "coal to electricity" campaign in the Beijing-Tianjin-Hebei region, might obtain net social benefits but exhibit a sizeable regional variation and inequality[16]. In addition, contemporary energy transition, such as increasing natural gas consumption, could effectively mitigate energy poverty among rural residents[17]. However, there are still considerable inequalities in energy consumption among rural households, particularly with regard to the usage of liquid petroleum gas (LPG), whereas the inequality in household electricity consumption is relatively more equitable[4]. Meanwhile, some scholars also estimated the energy transition potential by using bottom-up frameworks in an ex-ante way and proposed measures contributing to energy conservation and emission reduction, such as limiting the growth of floor space, promoting low-carbon electricity and high-efficiency renewable energy[18–22].

To summarize, previous studies mainly focused on the attributable air quality and health impacts associated with the historical residential energy transition, while some investigated the future energy conservation potential. However, few studies have taken a holistic look at rural energy transition pathways toward deep decarbonization and the spatial distribution of associated transformation costs versus air quality-related health benefits across the sub-national regions. Hence, it is unclear what the cost-effective contributions of and economic burdens are on the rural households under the ambitious dual national targets of carbon neutrality and air quality standard, which tend to overlook relatively lower-income rural residents (with a lower per capita disposable income than the urban residents), with notable regional heterogeneities and limited resources[23,24].

Here, focusing on RCH that account for over 80% of rural residential energy consumption[4], this study investigates how rural contemporary energy transition could contribute to China's carbon neutrality target and at what transformation costs. Meanwhile, we also attempt to uncover whether the air quality improvement-related health co-benefits could offset the transformation costs at the subnational level. To address these multi-facet questions and fill the knowledge gaps of previous studies[25,26], we propose an integrated assessment framework, which consists of a multi-province bottom-up energy system optimization model (IMED|TEC) well-calibrated with several representative nationwide residential energy surveys[3,4,11,27], an air quality model (GAINS) and a health impact assessment model (IMED|HEL). We construct two representative scenarios: baseline scenario (BaU) and carbon neutrality scenario (CNS). Additionally, a systematic sensitivity analysis has been conducted on the RCH system cost, energy consumption, emissions, air quality and associated health impacts to the changes in several vital parameters. Details regarding our analytical approach can be found in the Methods section, and additional information is provided in Part A and C of the Supplementary Information, including the key model parameters outlined in Part A.

## Results
### Energy consumption and its provincial disparity
Total energy consumption in BaU and CNS scenarios shows decreasing trends from 2014 to 2060, closely related to the decreased rural people due to rapid urbanization[28]. Even in BaU, the total energy consumption for RCH would almost halve from 151 Mtoe in 2014 (88 Mtoe for cooking and 63 Mtoe for heating) to 79 Mtoe in 2060 (Fig. 1a), a decrease rate of 47% that coincides with China's projected rural people reduction rate of 48% in the same period under the well-established shared socioeconomic pathways (SSP) 2[28]. This phenomenon reflects the underlying trends that potential energy consumption increase due to living standards enhancement could be almost offset by autonomous technological efficiency improvement and demographic change. More notably, it would decline to only 17% of the 2014

level in CNS in 2060, primarily because of the additional energy efficiency improvement for achieving carbon neutrality. Compared with BaU, the energy consumption in CNS for cooking decreased by 23 Mtoe in 2030 and 25 Mtoe in 2060. Conversely, the reduction amount is more dramatic for heating service in 2060 (28 Mtoe, Fig. 1b). From the perspective of energy types, biomass (wood and crop residues) consumption in 2060 still takes up 35% of the total energy use in BaU, not much reduction from 63% in 2014. By contrast, due to low carbon development and the ban on new biomass devices after 2030, the share of biomass and coal would plunge in CNS (Fig. 1a). Instead, contemporary energy consumption triples from 2014 to 2060 and the dominant energy in the rural area switches to electricity, of which 14.2 Mtoe is consumed for cooking (meeting 94% demand) and 4.3 Mtoe is consumed for heating (meeting 72% demand) in 2060. Noticeably, a two-stage energy transition from coal and biomass to contemporary energy could be observed, in which NG/LPG (natural gas/liquefied petroleum gas) replaces coal and biomass as the transitional fuel before 2040 and electricity will gradually become the dominant energy carrier in meeting the carbon neutrality target (Fig. 1b). Moreover, sensitivity analysis shows that a 1% change in any single parameter caused less than 0.97% and 0.01% change in national electricity and NG/LPG use in 2060, respectively. The detailed results are shown in Supplementary Fig. 5 and Supplementary Data 3.

From the perspective of regional disparity, rural energy consumption would decrease collectively in all provinces in both scenarios from 2014 to 2060, due to sustained service demand reduction (Table 1) and continuous energy efficiency improvement. Historically, since biomass fuels such as firewood and straw were free to collect and coal was cheap, considerable amounts of traditional biomass and coal were consumed in most provinces except for well-developed coastal areas. For instance, in the southwestern and northeastern provinces of Liaoning, Jilin, Guangxi and Sichuan, biomass accounted for 73%, 68%, 91% and 79% of their total rural energy consumption in 2014, respectively (Fig. 2a). Coal was also the vital energy in the main coal-producing provinces such as Shanxi and Guizhou, accounting for about 50% of their total energy in 2014. In addition, total energy consumption in the northern (Northern regions in China are shown in Supplementary Table 9) rural residential sector was higher than that in the southern areas in 2014, mainly due to a less efficient energy mix of solid fuels and higher demand for heating in the winter season. In BaU, coal and contemporary energy will gradually replace traditional biomass as the primary energy in northern China and southeast coastal areas in 2060, respectively (Fig. 2b). Ultimately, the energy consumption for RCH in CNS is projected to substantially decrease, while contemporary energy shares will increase to over 82% in all provinces in 2060, much higher than that in BaU (Fig. 2a–c). Moreover, the widespread penetration of advanced technologies narrowed the efficiency gaps between North and South China. The gravity center of rural energy consumption will shift from the north to the south in CNS.

The energy transition from coal and biomass toward contemporary energy ameliorates the energy poverty of rural residents, albeit leading to increased inequality of access to contemporary energy among provinces. Firstly, the contemporary energy share of almost all provinces would reach more than 90% in 2060 in CNS, a considerable increase compared with that in 2014 (Fig. 2a, c, e.g., nationwide contemporary energy share was only 6%) and BaU in 2060 (Fig. 2b, c). Secondly, national per capita contemporary energy use is projected to reach 80 kg of oil equivalent (kgoe) in 2060 in CNS, a considerable increase compared with that in 2014 (14 kgoe) and BaU in 2060 (23 kgoe). We further research the regional difference between per capita total energy use and per capita contemporary energy use at the provincial level based on the Gini coefficient to measure the inequality of access to energy and energy poverty, respectively. Encouragingly, reaching carbon neutrality for the rural residential sector reduces the inequality of per capita total energy use in CNS

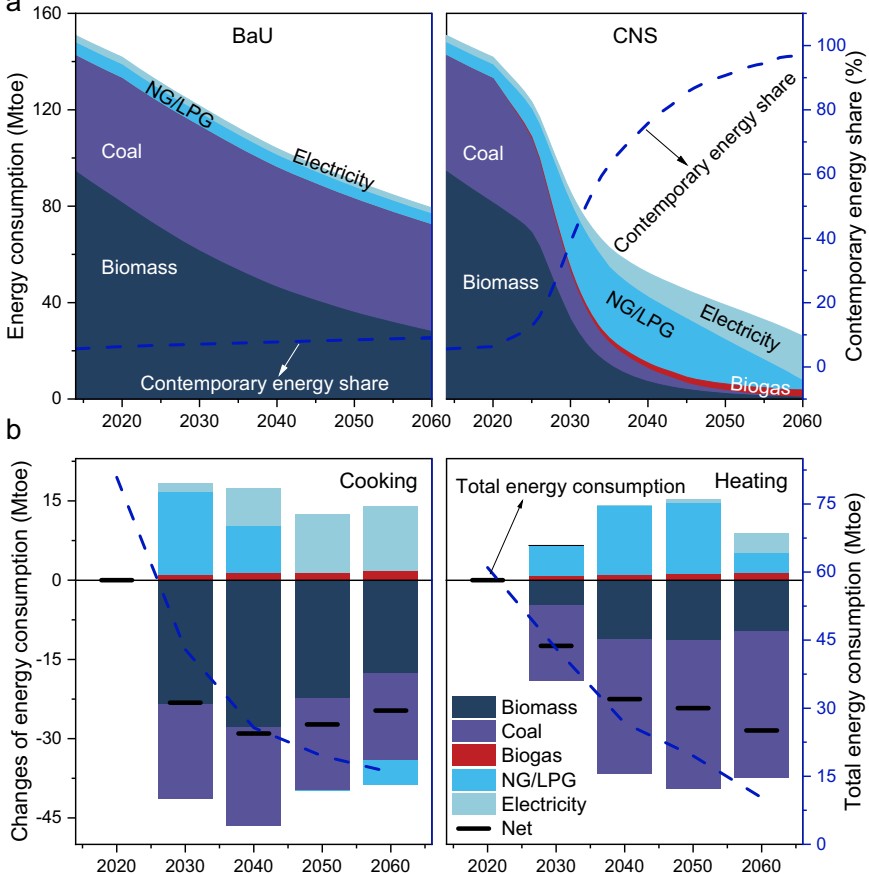

**Fig. 1 | National energy consumption and contemporary energy share for rural heating and cooking in China. a** Baseline scenario (BaU) on the left, and carbon neutrality scenario (CNS) on the right; **b** Changes in energy consumption in CNS relative to BaU, including total energy consumption for cooking (left) and heating (right). The black horizontal lines represent net total energy consumption change in CNS relative to BaU, while the blue dotted lines indicate total energy consumption in CNS.

relative to BaU. Nevertheless, it increases the inequality of per capita contemporary energy use (Fig. 2d). The Gini coefficient of per capita total energy use is expected to increase from 0.37 in 2014 to 0.41 in 2060, while that of per capita contemporary energy use is expected to decline slowly from 0.27 in 2014 to 0.25 in 2060 in BaU. However, in CNS, the Gini coefficient of per capita contemporary energy use is expected to peak at 0.36 in 2050 and then decrease to 0.33 in 2060. The widening and narrowing inequality fluctuations may originate from the regionally differentiated energy transition progress.

### Effects on emissions, air quality and health

Nationwide rural residential energy transition toward contemporary energy has widespread impacts on carbon and air pollutant emissions, ambient $PM_{2.5}$ concentrations and associated premature deaths. In 2014, the total $CO_2$ emissions from RCH were estimated to be 204 million metric tons (Mton), of which 110 Mton by heating and 94 Mton by cooking (Fig. 3a). Correspondingly, emissions of $SO_2$, $NO_x$ and $PM_{2.5}$ were 1.21, 0.58 and 2.56 Mton in 2014 (Fig. 3b–d),

respectively. In BaU, the total emissions of $CO_2$, $SO_2$, $NO_x$ and $PM_{2.5}$ would drop to 187, 1.08, 0.27, 1.24 Mton in 2060, respectively, which is 9%, 10%, 53% and 52% lower than those in 2014 thanks to the scale effects as mentioned above and structure effects of energy consumption. The declining magnitudes of $NO_x$ and $PM_{2.5}$ are much higher than $CO_2$ and $SO_2$ in BaU, owing to the gradual elimination of traditional biomass. Under the CNS scenario, the high-efficiency technologies and low-emission energy would substantially reduce $CO_2$ and air pollutant emissions. In 2060, emissions in CNS are further lowered by 95% for $CO_2$, 99% for $SO_2$, 97% for $NO_x$ and 99% for $PM_{2.5}$ (Fig. 3a–d) compared with the BaU scenario. Notably, sensitivity analysis shows that a 1% change in any single parameter caused less than 0.04% and 0.55% change in national $CO_2$ and $SO_2$ emissions in 2060, respectively (Supplementary Fig. 5 and Supplementary Data 3). We find that the substantial co-reduction effects of $CO_2$ and air pollutant emissions in CNS are mainly owing to the energy transition from biomass and coal toward electricity and NG/LPG, which are cleaner energy with lower emission factors of air pollutants. In addition, with electricity as the dominant energy in the rural residential sector for reaching carbon neutrality in 2060, direct emissions are transferred from end-use devices to upstream power plants. Thus, similar to findings in other sectors, decarbonization of power plants is crucial to rural residential life-cycle emission reductions. Emissions in the rural residential sector for cooking (blue) and heating (yellow) show an obvious decrease in CNS relative to BaU (Fig. 3a–d). Although the energy consumption of cooking is higher than that of heating, the latter will emit more $CO_2$ and $SO_2$ from 2014 to 2060 in BaU and contribute more emissions

**Table 1 | China's projected rural people, rural cooking and heating service demand under SSP2**

|  | 2014 | 2020 | 2030 | 2040 | 2050 | 2060 |
|---|---|---|---|---|---|---|
| Rural people (Billion people) | 0.61 | 0.57 | 0.48 | 0.41 | 0.36 | 0.32 |
| Cooking demand (Mtoe) | 24 | 23 | 19 | 17 | 14 | 13 |
| Heating demand (Mtoe) | 22 | 23 | 22 | 20 | 18 | 17 |

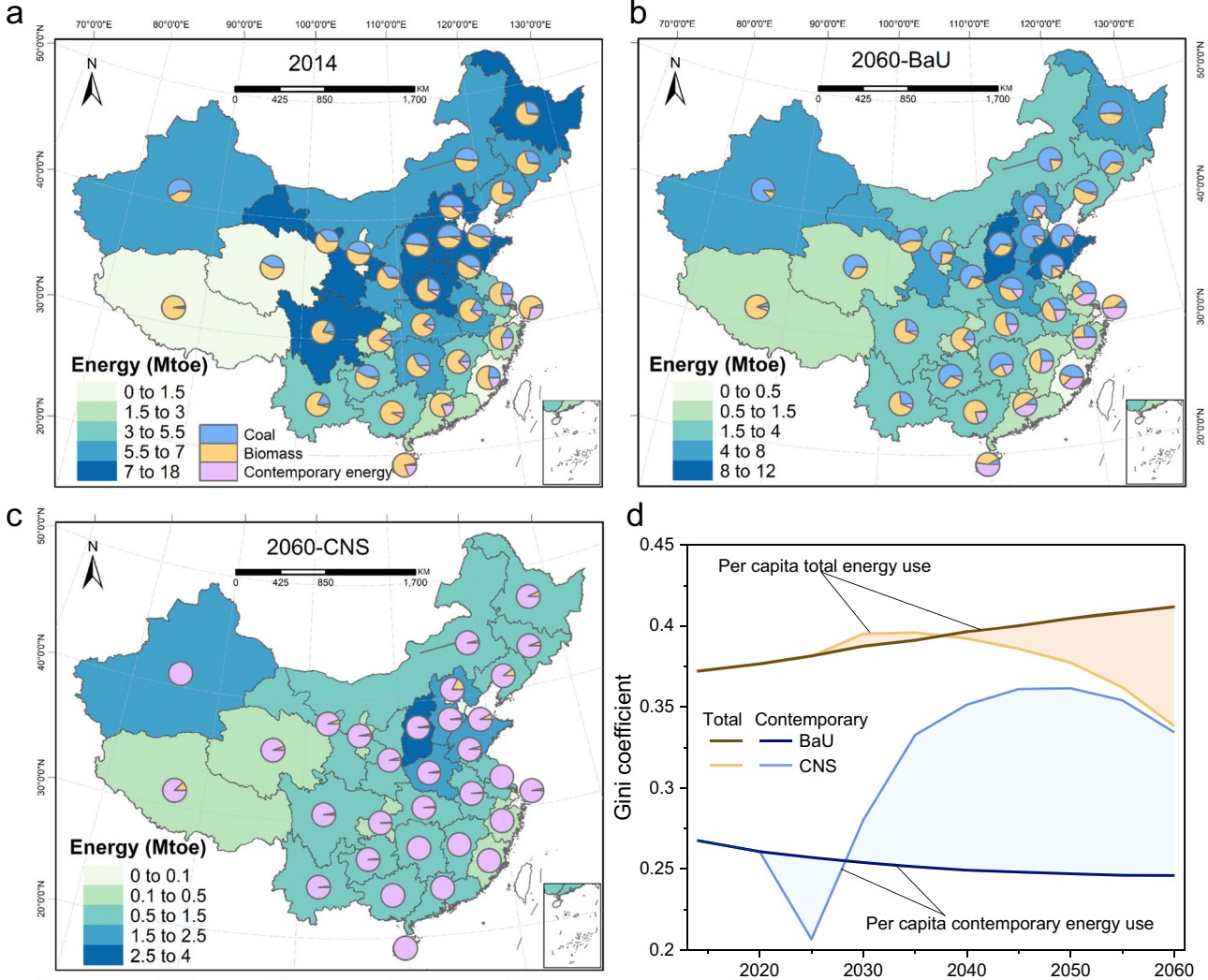

**Fig. 2 | Spatial distributions of rural cooking and heating energy consumption and its share. a** In 2014; **b** In baseline scenario (BaU) in 2060; **c** In carbon neutrality scenario (CNS) in 2060; **d** Gini index trends from 2014 to 2060 for per capita total energy use and per capita contemporary energy use. In (**a**–**c**), provincial regions are color-coded based on total energy use, and pi charts depict the energy mix. In (**d**), the dark and light lines represent the Gini coefficient in BaU and CNS, respectively, with shadows denoting changes in the Gini index. Base map data adapted from GS(2020)4619, http://bzdt.ch.mnr.gov.cn/.

reductions in CNS owing to higher coal substitution. Moreover, air pollutant emissions decrease precipitously by 2040 due to the phase-out of traditional biomass and scattered coal.

Furthermore, we apply GAINS and IMED|HEL models to simulate ambient $PM_{2.5}$ concentrations and associated premature deaths. It should be noted that the maximum time horizon of the GAINS model is by 2050. However, we still choose the GAINS model since it provides widely recognized dynamic emission pathways of non-residential sectors. Therefore, unlike the previous sections, air quality and health-related results in the following sections would only expand to 2050. The $PM_{2.5}$ concentrations in almost all provinces would fall due to the decarbonization of RCH across the country, especially in northern China, where $PM_{2.5}$ concentrations would reduce by 1.8–5.3 μg/m³ in 2035 (Supplementary Fig. 1a) and 1.9–5.4 μg/m³ in 2050 (Fig. 3e) in CNS relative to BaU (except for Inner Mongolia). Compared with northern China, the ambient air quality improvement in southern China is limited, especially in coastal areas where $PM_{2.5}$ concentration would drop by less than 1.5 μg/m³ in 2035 and 1.3 μg/m³ in 2050 (Fig. 3e, f and Supplementary Fig. 1). The regional differences indicate that the impacts of the energy transition (especially for rural heating) on air quality improvement in northern China are pivotal. This is because

intense air pollutant emissions related to solid household fuels used in Northern China contribute to server air pollution[29]. By the mid-century, the top five northern provinces witnessing the most notable air quality improvement in CNS relative to BaU would be Shanxi (5.4 μg/m³), Hebei (5.2 μg/m³), Shaanxi (4.6 μg/m³), Shandong (3.9 μg/m³) and Henan (3.8 μg/m³). Accordingly, ambient air quality improvement results in 69,500 and 75,500 avoided $PM_{2.5}$-associated premature deaths in 2035 and 2050 nationwide, respectively. Similar to air quality improvement, the health effects in northern China are projected to be considerably greater than those in southern China, with 63% and 69% of avoided $PM_{2.5}$-associated premature deaths occurring in north China in 2035 and 2050, respectively. Unexpectedly, the health benefits in eastern China are projected to be greater than those in western China (except for Sichuan) due to high population density. By the mid-century, the most notable health benefits would occur in Shandong, Henan and Hebei, with 9288, 9280 and 8014 avoided $PM_{2.5}$-associated premature deaths, respectively, where the reductions account for ~35% of national avoided premature deaths. Notably, the rural energy transition from coal and biomass to contemporary energy for reaching carbon neutrality has marked impacts on carbon emissions, air quality and human health. These impacts also show substantial regional

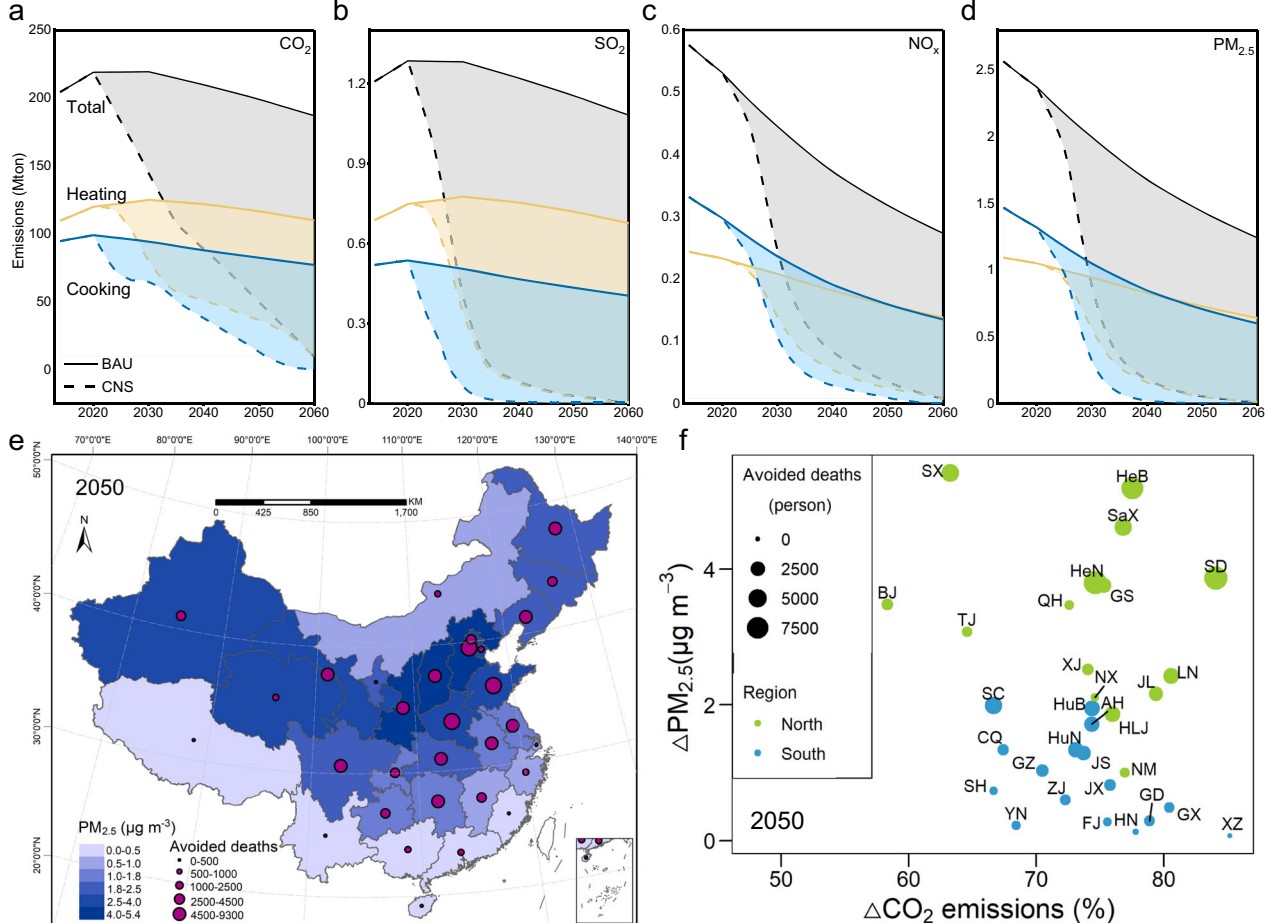

**Fig. 3 | Changes in emissions of China's rural cooking and heating. a** $CO_2$ emissions, **b** $SO_2$ emissions, **c** $NO_x$ emissions, and **d** $PM_{2.5}$ emissions; **e** Reduction in ambient $PM_{2.5}$ concentrations and avoided premature deaths in carbon neutrality scenario (CNS) relative to baseline scenario (BaU); **f** Regional disparity in carbon reduction, air quality improvement, and avoided $PM_{2.5}$-associated premature deaths in 2050. In (**a**–**d**), solid and dotted lines indicate emissions in BaU and CNS, respectively, with shadows representing emissions reductions in CNS relative to BaU. In (**e**), blue shades represent reductions in ambient $PM_{2.5}$ concentrations, and red dots indicate avoided $PM_{2.5}$-associated premature deaths in CNS relative to BaU. In (**f**), blue and dark green dots represent southern and northern provincial regions, respectively, with dot size reflecting provincial avoided $PM_{2.5}$-associated premature deaths. Base map data adapted from GS(2020)4619, http://bzdt.ch.mnr.gov.cn/.

disparity (Fig. 3f and Supplementary Fig. 1b). $CO_2$ emissions in most provinces will fall by over 40% by 2035 and 60% by 2050, except for Shanxi and Beijing, due to looser carbon constraints. In addition, the impacts of unit $CO_2$ emission reduction on air quality and health co-benefits show notable regional heterogeneity. For instance, regions with better background air quality would benefit less.

**Rural residential technology transformation and costs**
In parallel with the above energy transition and emission reduction, there will be substantial shifts in RCH technologies with considerable transformation costs to reach carbon neutrality. Rural residential technology transformation of cooking and heating at the national and provincial levels are detailed in Fig. 4a, b, Supplementary Figs. 1 and 2. In 2014, over 80% of cooking and heating service demand was met by technologies relying dominantly on traditional biomass and coal nationwide. It is mainly due to the cheap equipment and energy costs, with the total system costs, consisting of annualized capital costs and energy costs, being only 11.1 and 9.8 billion US$ for cooking and heating, respectively. From the perspective of regional disparity, in 2014, well-developed regions, such as coastal areas, were less dependent on the above traditional technologies. Especially, technologies for heating that rely on electricity play an essential role in meeting heating service in southern China. For instance, the technology shares

for Shanghai, Zhejiang and Fujian were ~50% in 2014. By contrast, there were no marked differences between northern and southern China in terms of technologies for cooking (Supplementary Fig. 2). Under the BaU scenario, the biomass cooking range switches to coal, LPG and electric cooking range. In contrast, biomass stoves mainly switch to coal stoves from 2014 to 2060 (Fig. 4a, b). Despite heating and cooking shifting to more expensive technologies, the cost of heating and cooking has not increased notably due to the decline of the rural people.

On the contrary, RCH technologies will undergo a two-stage dramatic shift under the CNS scenario. In the first stage by 2040, LPG cooking ranges and natural gas stoves would be the leading transitional technologies, meeting 47% of cooking service demand and 60% of heating service demand in 2040, respectively (Fig. 4a, b). Notably, the above transitional technologies would be predominantly in the northern provinces in 2040 due to relatively looser carbon constraints (Fig. 4c). On the contrary, almost all southern provinces use electric cooking range as the dominant technology for cooking constrained by strict carbon emission cap, while using air-to-air heat pump (AAHP) as the dominant technology for heating due to relatively faster autonomous technology improvement. It reflects that differential carbon constraints at the provincial level (related to the proportion of fossil energy in 2014) lead to faster technology transformation in southern

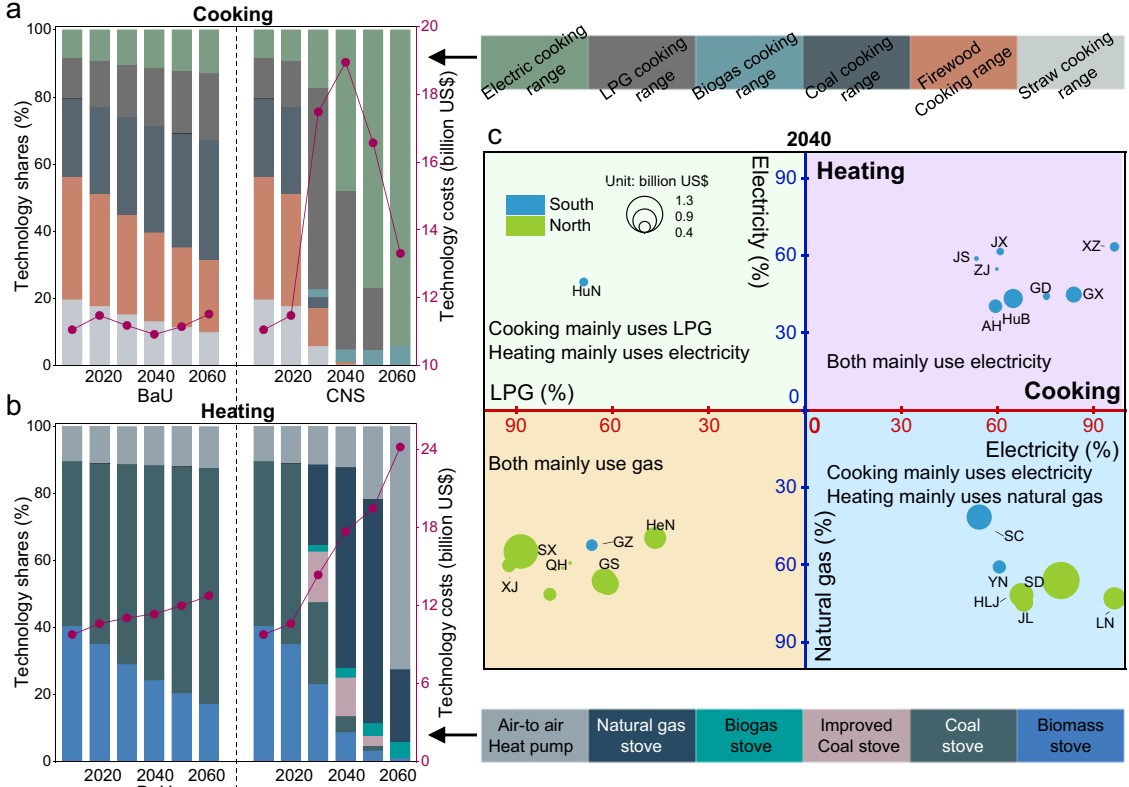

**Fig. 4 | Technology shares and annualized technology costs of rural cooking and heating. a** Cooking; **b** Heating; **c** Dominant contemporary technology and transformation costs for each province in 2040. In (**a**) and (**b**), the stacked area charts present technology shares, and dark red lines depict annualized technology costs. In (**c**), the dominant contemporary technology for heating is determined based on whether technologies relying on electricity or natural gas meet more heating service demand, and the same principle applies to cooking technologies as well. Blue and dark green dots indicate southern and northern provincial regions, respectively, with dot size indicating additional transformation costs. Note that economic costs and monetized benefits in our study are expressed at constant 2020 prices.

China in CNS. Nevertheless, northern provinces still pay more technology costs due to higher heating service demand and associated technology recruitment costs. Conversely, in the second stage by 2060, the electric cooking range and AAHP are the critical technologies for deep decarbonization, meeting 94% of cooking demand and 72% of heating demand in CNS, respectively. Notably, coal cooking ranges and coal stoves for heating will be nearly eliminated by 2040 under the CNS scenario, while biomass cooking ranges and stoves, which are least efficient but zero-carbon, are projected to exist in small quantities until 2060.

In addition, the total system cost of RCH would increase dramatically in CNS relative to BaU, especially for heating (Fig. 4a, b). After 2035, the system cost of cooking starts to fall to 13 billion US\$ in 2060, mainly due to the decline in the rural people. Conversely, the total cost of heating is projected to be as much as 24 billion US\$, primarily because of the extensive adoption of AAHP. Despite high efficiency, the unsubsidized capital cost of AAHP is ~3000 US\$ per unit[25]. Additionally, sensitivity analysis reveals that a 1% change in any single parameter caused less than 0.91% in total national system cost in 2060 (Supplementary Fig. 5 and Supplementary Data 3). Compared with BaU, the total additional transformation costs of deep decarbonization for RCH in CNS is 13 billion US\$ in 2060 (including 2 billion US\$ for cooking and 11 billion US\$ for heating), equivalent to 41 US\$ per capita[28] and 0.01% of GDP[30]. Considering the expected income growth, the abovementioned costs seem affordable for rural households. However, considerable additional transformation costs occur in Shandong, Heilongjiang, Shanxi and Hebei, with -1.7, -1.2, -1.1 and 1.1 billion US\$ (equivalent to -74, -150, -126 and 59 US\$ per capita) in

2060, respectively, altogether accounting for ~40% of national transformation costs (Supplementary Fig. 4). Thus, rural residents in the above regions bear substantial costs and may need subsidies or transfer payments.

## Cost-benefit analysis of RCH transition
Based on the localized value of statistical life (VSL) of each province (Supplementary Table 6), we estimate the monetized health benefits related to avoided $PM_{2.5}$-associated premature deaths in 2035 and 2050 at the medium, high and low levels. Furthermore, we compare the monetized health co-benefits of RCH transition with their additional transformation costs, which frames a cost-benefit analysis for RCH transition at the provincial level by 2035 ("Beautiful China" target year) and the mid-century (Fig. 5 and Supplementary Fig. 7). Results show that health benefits in the most provinces can offset transformation costs, except for certain southwestern provinces. Noticeably, the additional transformation costs and health co-benefits of RCH transition in northern China and in 2050 are greater than those in southern China and in 2035, mainly due to increasingly stringent carbon constraints and marked heating energy transition. When monetizing health benefits based on the medium VSL, most provincial regions, such as Shandong, Henan and Hebei (respective net benefits: 18.3, 11.0, 7.3 billion US\$ in 2035 and 31.6, 18.9, 14.0 billion US\$ in 2050), can entirely offset the transformation costs with health benefits, except for certain relatively lower-development regions (with a lower GDP per capita than the national average) like Tibet, where the negative net benefits are ~0.3 billion US\$ in both 2035 and 2050 (Fig. 5). Specially, the national benefit-cost ratio of RCH decarbonization in

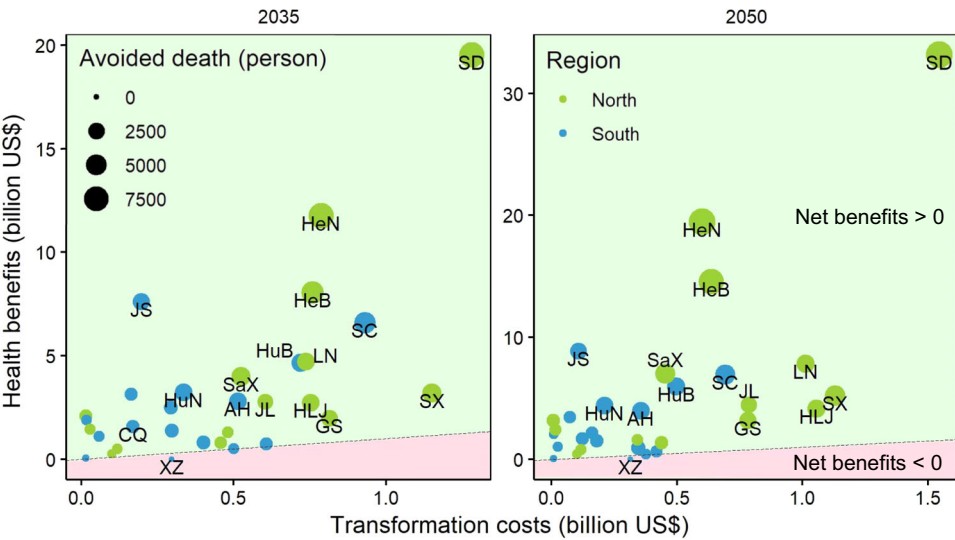

**Fig. 5 | Rural cooking and heating transformation costs and monetized health benefits related to avoided PM$_{2.5}$-associated premature deaths at the medium value of statistical life (VSL) level in 2035 and 2050.** Blue dots represent southern provincial regions, while dark green dots represent northern provincial regions. Provinces in the light green background area show positive net benefits (monetized health benefits minus transformation costs), whereas provinces in the pink background area exhibit negative net benefits.

2050 would reach 11.9. By contrast, when monetizing health benefits based on the low VSL, there would be 3 (Tibet, Yunnan, Guangxi) and 2 (Tibet, Yunnan) provinces where the health benefits could not offset the transformation costs in 2035 and 2050 (Supplementary Fig. 7), respectively. Note that the negative net benefits cannot outpace rural residents in lower-development regions to purchase contemporary energy equipment for decarbonizing cooking and heating. Therefore, targeted supporting measures such as subsidies are needed.

## Discussion

Our study shows that reaching carbon neutrality in the rural residential sector could bring considerable co-benefits by improving ambient air quality and human health, especially in northern China. China's annual average PM$_{2.5}$ concentration in 2021 is ~33 μg/m³, considerably above the updated WHO air quality guide (5 μg/m³). In 2050, each additional unit decrease in PM$_{2.5}$ concentrations would struggle as conventional pollution control measures run out of steam. Hence, reductions of 1.9–5.4 μg/m³ powered by RCH decarbonization efforts in almost all northern provinces matter a lot. Accordingly, decarbonization efforts in the residential sector alone could save about 75,500 PM$_{2.5}$-associated premature deaths nationwide in 2050, with 69% of the health benefits in northern China. Based on the extrapolation of historical technology mix trends at the provincial level in the baseline scenario (Supplementary Figs. 2 and 3), we assume that rural residents in northern China would still primarily use coal, leading to tremendous potential for improving human health. In addition, the number of national avoided premature deaths in 2050 in our study is only 36–56% of previous retrospective studies[10,26,31], which investigated the PM$_{2.5}$-associated health burden attributable to rural residential energy consumption in 2012 or 2015. The difference may be due to China's projected rural people will reduce by 42% during 2014–2050 under the SSP2 framework[28], and the heavy-polluting traditional biomass will gradually phase out by 2050 in BaU. Moreover, recent studies have examined the air quality and health impact of residential clean energy substitution in northern China through counterfactual scenarios[25], where the electric or NG/LPG equipment is uniformly applied following physical or engineering principles with insufficient economic considerations. In comparison, thanks to the cost optimization integrated model adopted in this study, we captured the under-studied policy issues by identifying a cost-effective transition pathway by the

mid-century in addition to uncovering the air quality and human health co-benefits.

Cumulative additional transformation costs for RCH transition from 2020 to 2060 are projected to be 430 billion US\$. Even with high transition costs, the additional transformation costs are projected to be offset by the health benefits in most provinces, except for several relatively lower-development regions with low background PM$_{2.5}$ concentrations. It is important to note that the VSL is even lower in the lower-development regions due to the innate characteristics of the value of the statistical life approach. For instance, Yunnan's medium VSL is projected to be only 16% of Shanghai's in 2050 (Supplementary Table 6). Although most provinces could gain net benefits, in theory, rural residential decarbonization is still challenging because rural residents do not prioritize the intangible chronic health benefits when compared with actual cash expenditure in upgrading household devices. Thus, it is necessary to improve rural residents' awareness of health benefits through information disclosure and education campaign to make the hidden benefits visible. In addition, our study suggests that the government should increase subsidies to purchase contemporary technologies, especially for rural residents in lower-development regions.

Promoting electric cooking range and AAHP is crucial for the deep decarbonization of RCH, meeting ~94% of cooking demand and ~72% of heating demand nationwide by 2060. Therefore, electricity consumption (~70% in 2060) is pivotal in meeting net-zero CO$_2$ emissions, which also transfers CO$_2$ emissions from rural household end uses to upstream power plants. The additional electricity demand from implementing electric cooking ranges and AAHPs (~580 kWh per capita in 2060) would challenge the decarbonization of the power sector. In addition, due to the decentralized characteristics of rural households, our study suggests that AAHP, a key decentralized heating technology, is projected to be widely integrated by 2060, which aligns with the current development trend of decentralized heating[19]. However, the heating effect of AAHP in winter is affected by frost and low ambient temperature, and its widespread adoption in cold and frosting regions is still in question. Nevertheless, the efficiency of AAHP is still 60–200% higher than resistance heaters[25], which is one of the heating options considered in IMED|TEC but not selected due to inferior cost competitiveness. In addition, more measures are used to overcome the shortcomings of AAHP, such as the quasi-two-stage compression heat

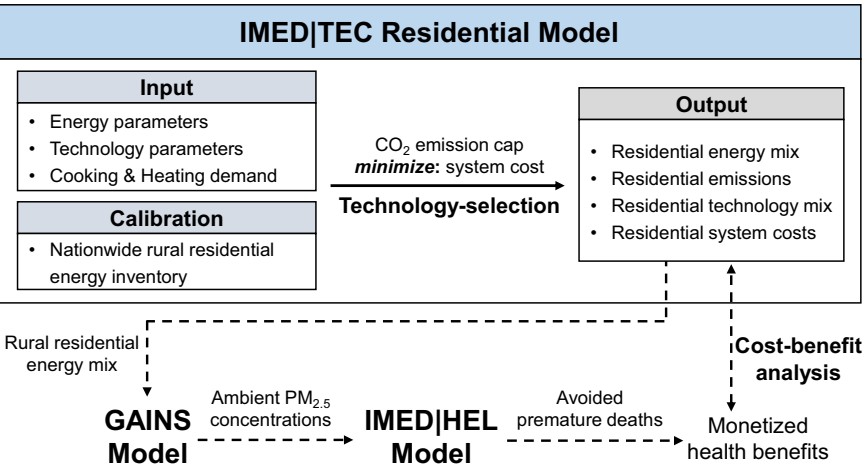

**Fig. 6 | Integrated modeling assessment framework.** IMED|TEC residential model is a multi-province bottom-up energy system optimization model, GAINS model functions as an air quality model and IMED|HEL model serves as a health impact assessment model. The models are interconnected through soft-linking: IMED|TEC residential model provides provincial rural residential energy mix (including energy consumption and structure) inputs to GAINS model, which, in turn, supplies ambient $PM_{2.5}$ concentrations to IMED|HEL model. By comparing the monetized health benefits derived from avoided premature deaths with the additional transformation costs, a cost-benefit analysis is conducted.

pump cycle[32,33], multi-source coupled cycle with a two-stage compression[34], hot air defrosting and liquid refrigerant defrosting technologies[35].

Rural residents in almost all regions switch toward contemporary energy as their primary fuel to achieve deep decarbonization, considerably improving energy poverty but increasing inequality of per capita contemporary energy use. Notably, the regional difference in per capita contemporary energy use is projected to widen under the carbon neutrality target, giving rise to varied rates of improvement in energy poverty across provinces. Since the per capita contemporary energy use in most provinces is close to the per capita total energy use in 2060, the regional difference in per capita contemporary energy use mainly comes from the regional disparity in per capita heating and cooking service demand related to regional socioeconomic development and climatic conditions. Therefore, measures to reduce economic development differences among regions, such as financial support for lower-development regions, are conducive to reducing the inequality of energy poverty, especially for the regional disparity in per capita contemporary energy use.

Sensitivity analysis shows that rural socioeconomic development-related cooking and heating demand have the most notable influence on RCH energy use, emissions and system cost. For instance, with a more people and lower income in SSP3, a 10% decrease in contemporary technology efficiency, 20% increase in the capital cost of contemporary technologies and contemporary energy prices, the system cost would be the highest (47.42% higher than that in CNS). Nevertheless, even though the alternative quantitative impacts differ from BaU and CNS, our conclusion that the health benefits of the rural contemporary energy transition to carbon neutrality outpace the transformation costs remarkably is plausible.

Our study still has some limitations and uncertainties. First, we chose 2014 as the base year because it was the most recent year when the provincial rural residential energy consumption survey data is publicly available. Without considering the clean heating policy gradually implemented in northern China after 2017, the contemporary energy share in Beijing, Tianjin, Hebei, Shandong and Shanxi in the baseline scenario may be underestimated, which leads to the overestimation of residential costs and health benefits in the above regions. Second, residential electricity is assumed to be carbon-free, although it may transfer $CO_2$ and air pollutants emissions to the power generation sector. In the future work, we will further couple the residential and

power generation sectors and explicitly consider the decarbonization pathways of the latter. Third, our study only considers the ambient $PM_{2.5}$-associated health burden attributed to rural residential energy consumption but does not consider indoor air pollution, which is thought to impact the health of rural residents considerably[6].

## Methods

Our study employs an integrated modeling assessment framework that combines the IMED|TEC (Integrated Model of Energy, Environment and Economy for Sustainable Development | Technology) model, the GAINS (Greenhouse Gas-Air Pollution Interactions and Synergies) model and the IMED|HEL (IMED| Health) model. Within this framework, we conducted two representative scenarios (baseline and carbon neutrality) as well as a comprehensive sensitivity analysis to explore the transition pathways towards carbon neutrality of RCH. More detailed information regarding these analyses can be found in the subsequent sections and Supplementary Information. The flowchart depicting the framework is presented in Fig. 6.

### IMED|TEC and Gini coefficient

The residential module of IMED|TEC focuses on the bottom-up technology-selection process with the least energy system cost in the residential sector, which allows for simulating the energy flows from energy sources to end-use services (e.g., cooking and heating) linked by different technologies. Specifically, there are six types of technologies for cooking and eight for heating, and the major energy types include traditional biomass, coal, biogas, NG/LPG and electricity (Supplementary Table 1). In accordance with previous studies, we define electricity, NG/LPG and biogas as contemporary energy[36]. Essentially, IMED|TEC is formulated as a linear program and the objective function for technology selection is based on the least total system cost (Eq. 1) and to meet the end-use service demand (Table 1), which is projected based on a statistical relationship between historical service demand and GDP per capita. Besides, IMED|TEC considers the dynamic balance of technology stock capacity, technology share constraints, $CO_2$ and air pollutants constraints under given assumptions[37–39]. Notably, the updated and provincial IMED|TEC model has been well calibrated with several representative nationwide residential energy surveys[3,4,11,27], and the energy consumption and air pollutant emissions in the model in the base year are well comparable to previous studies[3,4] (Supplementary Table 7). Comprehensive

information regarding the residential module of IMED|TEC and the primary datasets employed in our analysis can be found in Part A of the Supplementary Information.

$$TSC = \sum_t FI_t \cdot \frac{\alpha \cdot (1+\alpha)^{T_t}}{(1+\alpha)^{T_t} - 1} + \sum_t OMC_t + \sum_t EC_t + \sum_e E_e TaxE_e$$
$$+ \sum_g Q_g TaxG_g \qquad (1)$$

Where, $TSC$ represents the total energy system cost of the residential sector; $FI_t$ represents the fixed investments of technology $t$; $T_t$ represents the operating life of technology $t$; $\alpha$ represents the private discount rate (15%) for the residential sector[40,41]; $OMC_t$ represents the annual operation and maintenance costs of technology $t$; $EC_t$ represents the annual energy costs consumed by technology $t$; $E_e$ represents the consumption of energy $e$; $Q_g$ represents the emissions of $CO_2$ or air pollutants $g$; $TaxE_e$ and $TaxG_g$ represent annual energy tax for energy $e$ and annual emissions tax for gas $g$, respectively.

In our study, the key outputs of IMED|TEC include energy use, technology mix, emissions and associated system costs of RCH at the provincial level (Fig. 6). By comparing the system costs in different scenarios, we calculate the annualized transformation costs of RCH reaching carbon neutrality, including annualized capital costs and energy costs of various cooking and heating options in the rural residential sector. Capital costs are what households pay to purchase the devices. Furthermore, the annualized capital cost is calculated based on the expected lifespan of the devices and the above private discount rate for the residential sector. Annualized energy costs are what households pay for commercial fuels for cooking and heating each year.

We utilize the IMED|TEC model outputs to derive the Gini coefficient, enabling us to assess the disparity in rural residential energy use among provinces. To evaluate rural household energy poverty, we employ the energy development index (EDI) developed by the International Energy Agency[42]. The EDI takes into account the factors, including the contemporary energy share and per capita contemporary energy use. Specifically, we calculate the per capita total energy use and per capita contemporary energy use using provincial energy data from IMED|TEC along with people statistics under SSP2[28]. The Gini coefficients for per capita total energy use or per capita contemporary energy use are calculated as shown in Eq. 2 and then utilized to examine the disparity across provinces. Originally, Gini is a person-based coefficient like energy use per capita. Still, some studies also use the region-based index. For instance, ref. 43. conducted a literature review on the research which uses the Gini coefficient to analyze the inequality of regional carbon emissions.

$$Gini = \frac{1}{2n^2\mu} \sum_{i=1}^{n} \sum_{j=1}^{n} \left| EnPer_i - EnPer_j \right| \qquad (2)$$

Where, $n$ represents 31 provincial administrative regions in China (Hong Kong, Macau and Taiwan regions are not included due to lack of data); $i$ and $j$ represent two different provinces in China; $|EnPer_i - EnPer_j|$ represents all possible pair-wise of per capita total or contemporary energy use.

## GAINS model
GAINS model is an integrated energy-technology and air quality model developed by IIASA, which has highly integrated modules of the technology module, cost module, emissions module and air quality module. The GAINS model has been widely used to explore cost-effective multisectoral pollution control strategies for specific carbon or air quality targets, which is intensively documented in previous studies[44,45]. In this study, we focused on 31 provinces in China based on the GAINS-East Asia version, evaluating the air pollutant emissions and air quality impact of the rural residential sector under carbon and energy transformation targets. The input data is different energy consumption data at the provincial level from the IMED|TEC model. Since GAINS could only simulate air quality by 2050, we focus on 2035 and 2050, which correspond to the "Beautiful China" target year and is close to the "carbon neutrality" target year, respectively. Detailed information about the GAINS model is provided in Part A of the Supplementary Information.

## IMED|HEL model
IMED|HEL model includes two modules of health impact and monetization analysis, which calculates the burden of disease and the economic burden of health impacts related to $PM_{2.5}$ concentrations, respectively. This model has been applied widely and documented well in different scales of policy research[46-48], whose input data includes exposure or concentration levels of air pollution, the exposed population and exposure-response functions (ERFs) from the latest Global Exposure Mortality Model[49]. This model aims to quantify the number of $PM_{2.5}$-associated health co-benefits triggered by the energy transition among rural residents at the provincial level, such as $PM_{2.5}$-associated premature death. Detailed information pertaining to the IMED|HEL model is presented in Part A of the Supplementary Information.

## Scenario description
For simplicity but insightfulness, two representative scenarios, baseline scenario (BaU) and carbon neutrality scenario (CNS), are set up to illustrate the transition pathways of RCH toward carbon neutrality at the provincial level. BaU is the reference scenario without carbon emissions constraints, in which RCH would undergo a slow technology switch from 2015 to 2060, extrapolated from historical trends (1995–2014). We consider that the technology transformation process in BaU is mainly driven spontaneously by socioeconomic development, following energy ladder theory[50] and energy stacking theory[51]. By contrast, under the CNS scenario, the direct $CO_2$ emissions cap from RCH for each province would reduce by 25–35% in 2035 and 90–95% in 2060, respectively, compared with the 2014 level, which follows the 1.5 °C target of China's building sector[52] and makes appropriate adjustments according to provincial socioeconomic development. Detailed information regarding carbon emission constraints in CNS can be found in Supplementary Table 8. Considering rural revitalization, we should ban new traditional biomass devices after 2030 in CNS.

## Sensitivity analysis
In addition to the above two core scenarios (BaU and CNS), we applied one-at-a-time and two-at-a-time methods to investigate the sensitivity impacts of rural residential energy use, system costs, emissions, attributable $PM_{2.5}$ concentrations and related health impacts to changes in key input variables, to assess the robustness of the related results. The sensitivity analysis was performed covering six groups of key input variables, including (1) different SSP, covering people, urbanization and GDP, related to different RCH demand at the provincial level; (2) higher or lower contemporary technology capital cost; (3) higher or lower contemporary technology efficiency; (4) higher or lower contemporary energy price; (5) different ERFs; (6) higher or lower VSL. Detailed information about the sensitivity analysis conducted and the full suite of input parameters utilized are described in Part C of the Supplementary Information. Supplementary Data 1 and Supplementary Data 2 present the sensitivity scenarios based on the one-at-a-time and two-at-a-time methods, respectively. Furthermore, detailed results of the sensitivity scenarios based on the one-at-a-time and two-at-a-time methods are provided in Supplementary Data 3 and Supplementary Data 4, respectively.

## Reporting summary

Further information on research design is available in the Nature Portfolio Reporting Summary linked to this article.

## Data availability

Main data supporting the findings of this study are available within the Manuscript, Supplementary Information and Supplementary Data. Source data underlying figures in the Manuscript are provided within the Source Data. Source data are provided with this paper.

## Code availability

The code base of the IMED model have been developed at Peking University and related code used in this study is available from the corresponding authors and Hancheng Dai (dai.hancheng@pku.edu.cn) upon request. GAINS model is developed by International Institute of Applied Systems Analysis, which is available at https://gains.iiasa.ac.at/gains/.

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

## Acknowledgements
The study was supported by the National Natural Science Foundation of China (71903010, Y.X.; 72134006, Y.X.; 42077328, G.S.), Peking University-BHP Carbon and Climate Wei-Ming PhD Scholars (WM202305, T.M.), the China Postdoctoral Science Foundation (2022M720212, C.H.) and the Youth Academic Program in Area Studies of Peking University (7101602310, C.H.).

## Author contributions
T.M. and Y.X. designed the study and performed the data analysis; Y.X. provided supervision. T.M. and Y.X. wrote the paper; S.Z., Y.X., X.L. M.W. and K.W. provided comments. G.S. provided household energy data. C.H. provided support on visualization. T.M., Y.X. and Y.R.F. provide revision.

## Competing interests
The authors declare no competing interests.
