## [Peer Review File · Nature Communications]

Costs and health benefits of the rural energy transition to carbon neutrality in ChinaREVIEWER COMMENTS

Reviewer #1 (Remarks to the Author):

Modern energy transition of the rural residential sector is the key to achieving carbon neutrality. However, the energy transition pathway for carbon neutrality and the associated costs and benefits of the rural residential sector have not been systematically considered, leading to insufficient investment and attention in this key area. This study used an energy system optimization model to identify a cost-effective transition pathway for assessing the related air quality improvement and health impacts. It is a very timely and necessary study to build up the key carbon neutrality block at the regional level in China. The manuscript is clearly structured and well-written. The research is meaningful and could provide a reference for local and national governments for policy formulation regarding energy transition and carbon neutrality development for rural residential. This study is innovative by filling the above knowledge and policy gap and is worthy of recommendation for publication. I would recommend this manuscript as a candidate for publication in Nature Communications after some revisions.

1. The methodology part is very clear, but the validation process in the optimization model and the residential module of IMED|TEC should also be explained in the method or the appendix.
2. The provincial residential energy simulation model was calibrated to the base year 2014. Although the authors stated that it is the latest openly available data based on the household survey, 2014 is still a bit early from now. Therefore, I suggest the authors continue to calibrate the historical years incorporating the latest situation, such as 2015-2020. Accordingly, it will make the model parameters more accurate, and the model results will also be more reliable.
3. China is very wide, with significantly heterogeneous climate conditions and technological advancement among different regions or provinces. How does this study consider the regional heterogeneity of the efficiency of air-to-air heat pumps?
4. China's rural population is rapidly declining under urbanization. How does this study consider the impacts of urbanization on rural energy consumption and CO₂ emissions? For instance, as the rural population decreases, carbon emissions from the rural residential sector will decrease even without any low-carbon measures.
5. In the carbon neutrality scenario, what are the references for carbon caps for rural cooking and heating? It is recommended that scenarios be constructed based on the per capita carbon cap.
6. As one of the few provincial-level studies focusing on rural residential low-carbon transition, how does this study consider provincial disparities in rural heating and cooking demand? In particular, heat demand might be quite different in different provinces.
7. Provinces with different social-economic development might have different energy demands. How does this study consider the changing trends in the rural energy transition in the BaU scenario in different provinces, considering economic development and regional differences in rural areas?
8. Quantitative assessment of air quality and public health impacts of sectoral energy transitions is a recent research hotspot. Please compare the air and health impacts with recent relevant studies to verify the validity of the results.
9. It sounds inspiring that monetized health benefits in most provincial regions can offset the transformation costs. How do you estimate the VSL in different provinces? What are the costs of transformation included in this study?
10. As a forward-looking simulation, the authors are also recommended to conduct a sensitivity analysis in addition to the central scenario in the main text.
11. Please unify the text fonts in images, such as Figure 5 and others.
12. Please increase the font size in Figure 3a-3d to match the font size of the other figures.
13. Figure 6 needs improvement, as its meaning is not very clear.

Reviewer #2 (Remarks to the Author):

Summary of review: The paper captures a very important but often under studied policy issue: how rural population achieve clean energy transition to achieve carbon neutrality and what are the costs

around it. The energy-air pollution-human health modeling framework is comprehensive, and the results are insightful and policy relevant.

A few questions and comments for the paper to consider:

1. The analysis relies on a few key assumptions: rural population, rural household energy demand, and future energy prices. I would suggest the authors move some of the key assumption into the main text, and state the rationale for those assumptions clearly. Other assumptions could be included in the SI.
2. The paper designed two scenarios: BaU and CNS. CNS makes sense as it is a stated policy scenario. However, BaU assumes "extrapolated from historical trends (1995-2014)" which probably needs better justification.
3. Does the paper consider "stock turnover" of the cooking and heating technologies? Not sure where the technology structural changes come from. Are they part of modeling results? Or are they assumptions the authors construct? If they are the latter, is there any research to support those assumptions?
4. I do not see any in-depth discussions on how the results are sensitive to some key assumptions, such as urbanization, fuel structure, fuel prices, VSL, etc.
5. Other minor suggestions:
 - a. Ln41, better to give a rural population number in the same year.
 - b. Try to avoid using "Without a doubt (Ln49)".
 - c. The main text needs to better cite appendix tables and assumptions.

Reviewer #3 (Remarks to the Author):

This is a very interesting and meaningful topic, focusing on the energy transition in the rural area, with useful discussions on energy poverty, inequality, health impact. I have some comments as below.

1. In the Introduction section, are you suggesting that China's rural residential energy use is accounting for "4.7% of China's total energy consumption" but carbon emissions from the rural residential sector is accounting for "42.1% of China's total residential energy-related CO₂ emissions"? Why not show the proportion of "energy use in China's rural residential sector" in "energy use in China residential sector"? So that would help us understand the difference in carbon intensity between China's rural area and China as a whole.
2. In the Introduction section, why "switching rural residential energy toward modern energy is pivotal to simultaneously attain multi Sustainable Development Goals (SDGs) of rural revitalization (SDG 1, SDG 10)"? it doesn't seem very straight forward to me, and you also say the energy transition is "economic burdens" in page 3.
3. In the Introduction section, data is a bit out of date. Many pollution related data is date back to 2012, more than 10 years ago. Would be great to provide an update.
4. Methodology: for the residential module of IMED|TEC model, have you considered the reduction in costs brought about by technological progress? And in particular for electricity, it is more complicated as it will be determined by how the electricity is generated (i.e. the power generation mix). Has the model considered different technologies for electricity generation?

Response to Comments

Manuscript ID: NCOMMS-22-50254

Title: Effective contribution of rural modern energy transition to carbon neutrality and human health with acceptable costs in China

Date of comments received: 2023/3/24

Revision due before: 2023/6/24

Dear Reviewers,

Thanks for reviewing our manuscript. We have made revisions point by point after carefully reviewing the comments. Please find our response to the individual comments (shown in **blue** text). When showing changes to the text (Manuscript and Supplementary Information), sentences/words are quoted in quotes and marked in **purple** in this response letter.

Response to Reviewers

Reviewer #1:

Modern energy transition of the rural residential sector is the key to achieving carbon neutrality. However, the energy transition pathway for carbon neutrality and the associated costs and benefits of the rural residential sector have not been systematically considered, leading to insufficient investment and attention in this key area. This study used an energy system optimization model to identify a cost-effective transition pathway for assessing the related air quality improvement and health impacts. It is a very timely and necessary study to build up the key carbon neutrality block at the regional level in China. The manuscript is clearly structured and well-written. The research is meaningful and could provide a reference for local and national governments for policy formulation regarding energy transition and carbon neutrality development for rural residential. This study is innovative by filling the above knowledge and policy gap and is worthy of recommendation for publication. I would recommend this manuscript as a candidate for publication in Nature Communications after some revisions.

Response:

Thank you for your valuable comments and positive evaluation. They are insightful and contribute to the improvement of our work.

1. The methodology part is very clear, but the validation process in the optimization model and the residential module of IMED|TEC should also be explained in the method and the appendix.

Response:

In this study, the residential module of IMED|TEC is a provincial bottom-up optimization model, and we ensure provincial energy use in the base year is consistent with the representative nationwide rural household energy database conducted by Tao et al. (Tao et al., 2018). Especially, the benchmark survey is a bottom-up Chinese rural residential energy consumption database, which includes provincial fuel consumption of rural cooking and heating (including straw, firewood, coal, biogas, NG/LPG and electricity). Constrained by considerable workload and investment in conducting a nationwide survey, the latest historical year in this database is 2014. Nevertheless, it is considered the most precise national rural household energy data, and was widely used in multiple recent high-impact studies (Tao et al., 2018; Shen et al., 2019; Zhu et al., 2019; Yun et al., 2020; Meng et al., 2021).

Based on the unique database, the provincial and national energy consumption of rural cooking and heating in 2014 in the IMED|TEC model have been calibrated to be well comparable with recent studies (Tao et al., 2018; Yun et al., 2020). Furthermore, we also validated the air pollutant emissions. The updated comparison of the estimated air pollutants emissions and energy consumption of rural cooking and heating in 2014 is shown in Supplementary Information - Table A.7.

We also calibrated the model results with another well-known but less-coverage rural household energy consumption survey (Chinese Residential Energy Consumption Survey (CRECS) 2013 launched by the Renmin University of China), which only covers 12 provinces in China. Two published key indicators of energy use per capita and share of biomass use are very close between our results and the findings based on CRECS 2013 (Zheng, 2015; Wu et al., 2017).

Based on your comments, we have revised the original Table A. 7 in the Supplementary

Information - Appendix. A as the new table-Table A.7 in the Supplementary Information-Appendix A.

Table A.7 Comparison of the estimated air pollutants emissions and energy consumption of rural cooking and heating in 2014 of this study and from the other literature.

	This study	Tao et al. (Tao et al., 2018) ^a	Yun et al.(Yun et al., 2020)	Wu et al. (Zheng, 2015;Wu et al., 2017) ^b
SO ₂ (Mton)	1.21	1.20	1.3	-
NO _x (Mton)	0.58	0.64	-	-
PM _{2.5} (Mton)	2.56	2.72	2.4	-
Energy use (Mtoe)	151	184	148	-
Energy use per capita (kgoe)	246	-	-	236
Share of biomass use (%)	63	67	64	61

^a. The data is for the year 2012. The energy use data is estimated based on the sum of various fuel use. Note that rural residential energy consumption for cooking and heating, estimated by Tao et al. decreased by 60 Mtoe between 2007 and 2012.

^b. The data is for the year 2013. The energy use per capita is estimated based on the total rural household energy consumption per capita (267 kgoe) and the share of cooking and heating energy use (88%). The share of biomass use takes into account biogas, which refers to the proportion of total energy consumption, not the proportion of cooking and heating energy consumption.

Reference

1. Tao S, *et al.* Quantifying the rural residential energy transition in China from 1992 to 2012 through a representative national survey. *Nature Energy* **3**, 567-573 (2018).
2. Yun X, *et al.* Residential solid fuel emissions contribute significantly to air pollution and associated health impacts in China. *Science Advances* **6**, (2020).
3. Zhu X, *et al.* Stacked Use and Transition Trends of Rural Household Energy in Mainland China. *Environmental Science & Technology* **53**, 521-529 (2019).
4. Shen G, *et al.* Impacts of air pollutants from rural Chinese households under the rapid residential energy transition. *Nature Communications* **10**, 3405-3408 (2019).
5. Meng W, *et al.* Synergistic Health Benefits of Household Stove Upgrading and Energy Switching in Rural China. *Environmental Science & Technology* **55**, 14567-14575 (2021).
6. Wu S, *et al.*. Measurement of inequality using household energy consumption data in rural China. *Nature Energy* **2**, 795-803 (2017).
7. Zheng X. *China Household Energy Consumption Research Report*. Science Press (2015).

2. The provincial residential energy simulation model was calibrated to the base year 2014. Although the authors stated that it is the latest openly available data based the on household survey, 2014 is still a bit early from now. Therefore, I suggest the authors continue to calibrate the historical years incorporating the latest situation, such as 2015-2020. Accordingly, it will make the model parameters more accurate, and the model results will also be more reliable.

Response:

Many thanks for these comments.

As mentioned in the last response, constrained by the considerable workload and cost of conducting a nationwide survey for rural household energy use, there are no more recent national rural household energy surveys that can better represent the provincial level rural household energy consumption in China than we used in our study. This representative nationwide rural household energy database was conducted by Tao et al. (Tao et al., 2018) and the most recent historical data is updated to 2014. This dataset was widely used in several high-impact recent studies (Tao et al., 2018; Shen et al., 2019; Zhu et al., 2019; Yun et al., 2020; Meng et al., 2021) and is considered to be the most precise rural household energy data in China so far. Furthermore, we tried to extend the historical trend in the BaU and partly incorporated more recent policies in the CNS scenario to reflect the effects of the stated policies from 2014 to 2021. Therefore, we believe it is acceptable to calibrate the base year data 2014 associated with stated policies in the simulation years by 2021 in the model.

Additionally, clean heating policies are still not widely implemented in most rural areas of China, except for part of northern China, so it is relatively reasonable to set in the baseline scenario (BaU) where rural cooking and heating undergo a spontaneous but slow energy switching during 2015-2020, extrapolating from historical trends (1995-2014).

Nevertheless, our study inevitably ignores the implementation of Clean Heating Plan for Northern China in the Winter for 2017–2021, due to the data unavailability, leading to the limitation as stated in Line 397 in Manuscript: "First, we chose 2014 as the base year because it was the most recent year when the provincial rural residential energy consumption survey data is publicly available. Without considering the clean heating policy gradually implemented in northern China after 2017, the modern energy share in Beijing, Tianjin, Hebei, Shandong and Shanxi in the baseline scenario may be underestimated, which leads to the overestimation of health benefits in the above regions."

Reference

1. Tao S, *et al.* Quantifying the rural residential energy transition in China from 1992 to 2012 through a representative national survey. *Nature Energy* **3**, 567-573 (2018).
2. Yun X, *et al.* Residential solid fuel emissions contribute significantly to air pollution and associated health impacts in China. *Science Advances* **6**, (2020).
3. Zhu X, *et al.* Stacked Use and Transition Trends of Rural Household Energy in Mainland China. *Environmental Science & Technology* **53**, 521-529 (2019).
4. Shen G, *et al.* Impacts of air pollutants from rural Chinese households under the rapid residential energy transition. *Nature Communications* **10**, 3405-3408 (2019).
5. Meng W, *et al.* Synergistic Health Benefits of Household Stove Upgrading and Energy Switching in Rural China. *Environmental Science & Technology* **55**, 14567-14575 (2021).

3. China is very wide, with significantly heterogeneous climate conditions and technological advancement among different regions or provinces. How does this study consider the regional heterogeneity of the efficiency of air-to-air heat pumps?

Response:

Thank you for pointing out this important issue. In this revision, we considered the heterogeneous climate conditions and technological advancement among different provinces. Specially, we agree that the efficiency of air-to-air heat pumps (AAHP) is different among provinces due to climate conditions. In our revised manuscript, provincially adjusted efficiency parameters of AAHP have been involved in the modeling based on previous studies (Yang, 2018;Zhou et al., 2021), as shown in Eq.1.

$$\text{efficiency of AAHP} = 0.07 \times \text{ambient temperature } (^{\circ}\text{C}) + 2.69 \quad (1)$$

We collected the average ambient temperatures during the heating seasons from 2010-2017 (November, 2010-February, 2018) at the meteorological sites of the provincial capitals for calculating the efficiency of AAHP, as shown in Supplementary Information - Table A.2. After considering the regionally adjusted efficiency parameters of AAHP in the modeling, the fourth limitation of our study is solved so that we delete it.

Based on your comments, we have added the new table-Table A.2 in Supplementary Information- Appendix A.

Table A.2. Provincial efficiency of AAHP and ambient temperatures

Provinces/ municipalities/ autonomous regions	Average ambient temperatures during the heating season*	Efficiency of AAHP
Anhui	6.41	3.14
Beijing	0.4	2.72
Chongqing	10	3.39
Fujian	13.94	3.67
Gansu	-0.61	2.65
Guangdong	16.12	3.82
Guangxi	14.98	3.74
Guizhou	6.93	3.18
Hainan	20	4.09
Hebei	1.9	2.82
Heilongjiang	-12.33	1.83
Henan	4.37	3.00
Hubei	7.22	3.2
Hunan	8.55	3.29
Inner Mongolia	-6.47	2.24
Jiangsu	6.48	3.14
Jiangxi	9.18	3.33
Jilin	-9.86	2.00
Liaoning	-6.83	2.21
Ningxia	-2.84	2.49
Qinghai	-4.79	2.35
Shaanxi	3.51	2.94

Shandong	3.01	2.9
Shanghai	8.25	3.27
Shanxi	-1.23	2.6
Sichuan	8.78	3.3
Tianjin	0.41	2.72
Tibet	2.08	2.84
Xinjiang	-7.83	2.14
Yunnan	10.88	3.45
Zhejiang	8.22	3.27

*Average ambient temperatures during the heating seasons from 2010-2017 (November, 2010-February, 2018) come from the national meteorological observatory database. This database is managed by China Meteorological Data Service Center.

Reference

1. Yang X. Current situations and technical routes of rural clean heating (in Chinese). *The 14th session of Building Energy Efficiency Academic Week in Tsinghua University: Clean Heating Forum, Beijing*, (2018).
2. Zhou M, *et al.* Environmental benefits and household costs of clean heating options in northern China. *Nature Sustainability* **5**, 329-338 (2021).

4. China's rural population is rapidly declining under urbanization. How does this study consider the impacts of urbanization on rural energy consumption and CO₂ emissions? For instance, as the rural population decreases, carbon emissions from the rural residential sector will decrease even without any low-carbon measures.

Response:

Exactly, you are right, China is under fast urbanization and will continue in the future. This study considered the impacts of China's urbanization and rural population change on future energy consumption and CO₂ emissions. Table 1 shows China's projected urbanization rate and rural population by 2060 under the well-established SSP2 framework (Chen et al., 2020). We first estimated future cooking and heating demand per capita based on historical service demand per capita (dividing historical residential energy consumption by device efficiency) and projected GDP per capita. Further, the future service demand was estimated by multiplying the projected rural population by service demand per capita. We observed exactly the same trend as you mentioned: the decline in rural population reduces cooking and heating demand, further leading to lower rural energy consumption and CO₂ emissions even with a fixed energy mix.

We also added this explanation in Supplementary Information - Appendix A.1.1: “Therefore, projected rural population reduction reduces cooking and heating demand, further leading to lower rural energy consumption and CO₂ emissions even with fixed energy mix.”.

Table 1. Projected China's urbanization rate and rural population

	2014(NBSC, 2015)	2060(Chen et al., 2020).		
		SSP2	SSP1	SSP3

Urbanization (%)	55.3%	75.9%	80.0%	71.9%
rural population (billion)	0.61	0.32	0.25	0.4

Furthermore, we conducted a sensitivity analysis by considering alternative total population and urbanization rates under the SSP1 and SSP3 scenarios. The sensitivity analysis of rural socioeconomic development changing from SSP2 to SSP1 (SSP3) in BaU, the total energy consumption and CO₂ emissions of China's rural cooking and heating will decrease by 18.1% and 17.9% (increase 19.4% and 17.9%), respectively (seeing Supplementary Sheet. 3). **This sensitivity analysis is also mentioned in Supplementary Information - Appendix C.**

Reference

1. NBSC. *China Statistical Yearbook 2014*. (2015).
2. Chen Y, *et al.* Provincial and gridded population projection for China under shared socioeconomic pathways from 2010 to 2100. *Scientific Data* 7, 83 (2020).

5. In the carbon neutrality scenario, what are the references for carbon caps for rural cooking and heating? It is recommended that scenarios be constructed based on the per capita carbon cap.

Response:

We fully agree with your comment and have revised the carbon cap based on the per capita CO₂ emission for the carbon neutrality scenario (CNS). Our study referred to "China's New Growth Pathway: From the 14th Five-Year Plan to Carbon Neutrality" (EFC, 2020) on the emission reduction pathway for the building sector under the 1.5°C target. We made appropriate adjustments according to provincial socioeconomic development. The direct CO₂ emissions cap from rural cooking and heating (RCH) for each province would reduce by 25-35% in 2035 and 90-95% in 2060, respectively, compared with the 2014 level. The provincial CO₂ emissions from RCH in 2014 were calculated based on the provincial RCH energy use and carbon emission factors. In SSP2, China's rural population is projected to reduce by half from 2014 to 2060 under rapid urbanization in China (Chen et al., 2020), halving CO₂ emissions from rural cooking and heating even without any low-carbon measures. Therefore, the carbon cap of rural cooking and heating by 2060 with a 90-95% reduction in the per capita carbon emission compared with the 2014 level is a more plausible scenario for the rural residential sector achieving carbon neutrality. **We have revised the original Table A.8 in Supplementary Information - Appendix A as the new table - Table A.8 in Supplementary Information - Appendix A.**

Moreover, we have revised all the results based on the renewed carbon neutrality scenario, including associated figures and related contents in the Manuscript and Supplementary Information.

Table A.8 Provincial CO₂ cap of rural cooking and heating during 2020-2060 (Mton)

Region	Abbreviation	2020	2035	2060
Anhui	AH	4.905	2.391	0.242
Beijing	BJ	1.043	0.298	0.005
Chongqing	CQ	0.772	0.330	0.030
Fujian	FJ	1.246	0.498	0.025

Gansu	GS	14.255	6.937	0.610
Guangdong	GD	1.525	0.714	0.040
Guangxi	GX	0.781	0.405	0.044
Guizhou	GZ	8.215	4.780	0.508
Hainan	HN	0.076	0.042	0.005
Hebei	HeB	23.120	10.597	1.107
Heilongjiang	HLJ	8.813	4.947	0.407
Henan	HeN	11.420	5.528	0.566
Hubei	HuB	4.332	2.302	0.223
Hunan	HuN	7.956	3.967	0.397
Inner Mongolia	NM	12.689	4.939	0.224
Jilin	JS	7.978	4.174	0.350
Jiangsu	JX	3.802	1.651	0.093
Jiangxi	JL	2.325	1.174	0.122
Liaoning	LN	7.361	3.013	0.132
Ningxia	NX	2.437	1.395	0.156
Qinghai	QH	2.700	1.492	0.139
Shaanxi	SaX	9.712	4.600	0.421
Shandong	SD	22.991	9.648	0.517
Shanghai	SH	0.183	0.069	0.002
Shanxi	SX	31.102	18.528	1.786
Sichuan	SC	6.653	3.245	0.275
Tibet	TJ	0.178	0.093	0.009
Tianjin	XZ	1.119	0.404	0.011
Xinjiang	XJ	13.838	9.044	1.096
Yunnan	YN	2.947	1.582	0.148
Zhejiang	ZJ	2.452	1.079	0.052

Reference

1. EFC. Synthesis Report 2020 on China's Carbon Neutrality: China's New Growth Pathway: from the 14th Five Year Plan to Carbon Neutrality. Energy Foundation China, Beijing, China. (2020).
2. Chen Y, *et al.* Provincial and gridded population projection for China under shared socioeconomic pathways from 2010 to 2100. *Scientific Data* 7, 83 (2020).

6. As one of the few provincial-level studies focusing on rural residential low-carbon transition, how does this study consider provincial disparities in rural heating and cooking demand? In particular, heat demand might be quite different in different provinces.

Response:

We have considered the provincial disparities in our analysis regarding disparities in rural heating and cooking demand. Specifically, we analyzed the historical year (2005-2014) of per capita heating and cooking demand (dividing historical residential energy use by device efficiency) at the provincial level to predict future heating and cooking demand. We found that during 2005-2014, the per capita cooking demand in almost all regions and heating demand per capita in the

southern regions did not significantly change. In contrast, the heating demand per capita in the northern regions was positively correlated with GDP per capita. Therefore, we fitted the heating demand per capita (HD_{per}) with the GDP per capita (GDP_{per}) to project the disparity of heating demand per capita among provinces (Eq. 2) and listed the related regression coefficients in Table 2.

$$HD_{per} = \alpha \times \ln(GDP_{per}) + \beta \quad (2)$$

Where, HD_{per} represents the heating demand per capita; GDP_{per} represents the GDP per capita; α and β represent the regression parameters, which are listed in Table 2.

Table 2. Regression parameters of Eq. (2)

	α	β	t Stat	adjusted R^2
Guizhou	52.58	-166.00	8.65	0.89
Henan	44.02	-133.68	3.65	0.58
Inner Mongolia	323.67	-2128.55	7.95	0.87
Jilin	251.38	-1503.94	4.83	0.71
Liaoning	334.84	-2431.54	8.29	0.88
Ningxia	118.84	-398.35	3.31	0.53
Qinghai	122.01	-223.35	5.23	0.75
Shaanxi	163.82	-946.67	15.91	0.97
Shandong	250.81	-1905.84	5.69	0.78
Xinjiang	161.15	-870.61	5.00	0.73
Gansu	351.89	-2198.16	6.79	0.83
Hebei	211.57	-1406.67	2.88	0.45

Note: For the other regions, due to no significant changes in cooking or heating demand, we set their future service demand per capital fixed at the 2014 levels. Shanxi's historical per capita heating demand is too high due to the high accessibility of coal, so we set its future per capita heating demand fixed at the 2014 level considering the clean energy transition trends. Additionally, the adjusted R^2 of the fitted regression for Heilongjiang is small. Therefore, Jilin, which is similar to its climatic conditions and socioeconomic situation, is chosen instead.

Notably, the projected provincial service demand takes into account regional socioeconomic and climatic heterogeneity based on the above sub-national estimation approach. **The explanation is also shown in Supplementary Information Appendix A.1.1 as follows:**

“For certain regions, rural cooking or heating service demand can be calculated by multiplying the future rural population under shared socioeconomic pathways (SSP) and future annual service demand per capita. Future annual service demand per capita is estimated based on a statistical relationship between historical service demand per capita and GDP per capita, where historical service demand per capita is estimated by dividing historical residential energy consumption (2005-2014) by device efficiency. Specifically, it is found that during 2005-2014, the cooking demand per capita in almost all regions and heating demand per capita in the southern regions did not change significantly, whereas the heating demand per capita in most northern regions was positively correlated with GDP per capita. Therefore, we assume that the future cooking demand per capita in

all regions and heating demand per capita in the southern regions remain at the 2014 level, whereas the future heating demand per capita in certain northern regions is adjusted according to per capita GDP. Notably, the projected provincial service demand takes into account regional socioeconomic and climatic heterogeneity based on the above sub-national estimation approach.”

7. Provinces with different social-economic development might have different energy demands. How does this study consider the changing trends in the rural energy transition in the BaU scenario in different provinces, considering economic development and regional differences in rural areas?

Response:

Based on your comments, we have added the explanation in Supplementary Information - Appendix A.1.1, as follows:

“Notably, the projected provincial service demand could capture the regional socioeconomic and climatic heterogeneity based on the above sub-national estimation approaches.”

Provinces with different socioeconomic and climatic characteristics have different cooking and heating demand. Based on the historical provincial energy use of rural cooking and heating, we calculated the cooking and heating demand of each province by dividing historical residential energy use by device efficiency. Specifically, we analyzed the historical year (2005-2014) of per capita heating and cooking demand (dividing historical residential energy use by device efficiency) for different provinces, further to predict future data. We found that during 2005-2014, the cooking demand per capita in almost all regions and heating demand per capita in the southern regions did not significantly change, whereas the heating demand per capita in the northern regions was positively correlated with GDP per capita. Furthermore, we estimate the future rural cooking and heating demand based on the projected rural population (Chen et al., 2020) and GDP per capita (Mondal et al., 2021) at the provincial level under the SSP2 framework.

Notably, the historical heating and cooking demand per capita in 31 provinces are related to their socioeconomic and climatic characteristics. The SSP2 framework also provides future socioeconomic heterogeneity of the rural residential sector across provinces. Therefore, the provincial heterogeneity of the estimated future provincial cooking and heating demand is captured.

Reference

1. Chen Y, *et al.* Provincial and gridded population projection for China under shared socioeconomic pathways from 2010 to 2100. *Scientific Data* 7, 83 (2020).
2. Mondal SK, *et al.* Doubling of the population exposed to drought over South Asia: CMIP6 multi-model-based analysis. *The Science of the Total Environment* 771, 145186 (2021).

8. Quantitative assessment of air quality and public health impacts of sectoral energy transitions is a recent research hotspot. Please compare the air and health impacts with recent relevant studies to verify the validity of the results.

Response:

Our study found decarbonization of rural cooking and heating is projected to reduce PM_{2.5} concentrations by 1-5.4 µg/m³ at the provincial level and avoid 75,500 PM_{2.5}-associated premature

deaths nationwide in 2050. Yun et al.(Yun et al., 2020) found that the contribution of rural cooking and heating to the national rural and urban ambient PM_{2.5} concentration in 2014 was 3.9 and 4.9 µg/m³, respectively. Several recent studies have estimated that the PM_{2.5}-associated health burden attributable to rural residential energy in 2012(Shen et al., 2019;Zhao et al., 2019) or 2015(Lu et al., 2022) ranged from 135,000 to 210,000. Our results in 2050 are only 36%-56% of those studies in the 2010s. The reason is that China's projected rural population will reduce by 42% during 2014-2050 and the heavy-polluting traditional biomass will gradually phase out by 2050 in BaU. Therefore, considering both effects of rural population reduction and biomass phaseout, our estimated air quality and health impacts are reasonable in order of magnitude compared with recent studies. **The related comparison is shown in the Discussion in Manuscript as Line 338 :**

"In addition, the number of national avoided premature deaths in 2050 in our study is only 36-56% of previous retrospective studies (Shen et al., 2019;Zhao et al., 2019;Lu et al., 2022), which investigated the PM_{2.5}-associated health burden attributable to rural residential energy consumption in 2012 or 2015. The difference may be due to China's projected rural population will reduce by 42% during 2014-2050 under the SSP2 framework (Chen et al., 2020) and the heavy-polluting traditional biomass will gradually phase out by 2050 in BaU."

Reference

1. Yun X, et al. Residential solid fuel emissions contribute significantly to air pollution and associated health impacts in China. *Science Advances* 6, (2020).
2. Shen G, et al. Impacts of air pollutants from rural Chinese households under the rapid residential energy transition. *Nature Communications* 10, 3405-3408 (2019).
3. Zhao H, et al. Inequality of household consumption and air pollution-related deaths in China. *Nature Communications* 10, 4337-4339 (2019).
4. Lu C, et al. Reduced health burden and economic benefits of cleaner fuel usage from household energy consumption across rural and urban China. *Environmental Research Letters* 17, 14039 (2022).
5. Chen Y, *et al.* Provincial and gridded population projection for China under shared socioeconomic pathways from 2010 to 2100. *Scientific Data* 7, 83 (2020).

9. It sounds inspiring that monetized health benefits in most provincial regions can offset the transformation costs. How do you estimate the VSL in different provinces? What are the costs of transformation included in this study?

Response:

We adopt the willingness to pay method (WTP) to monetize the health benefits of air quality improvement. This study adopts the value of statistical life from Jin et al.(Jin and Zhang, 2018) and the elasticity coefficient to calculate the different VSL calibration values corresponding to different GDP per capita levels of China's provinces in 2035 and 2050. This adjustment approach of VSL is widely used in previous studies(Gao et al., 2021;Yin et al., 2021;Yan et al., 2022). **The provincial VSL values are listed in Supplementary Information - Table A.6, and a detailed description is shown in Supplementary Information - Appendix A.3.**

"We use the elasticity coefficient to calculate the different value of statistical life (VSL)

calibration values corresponding to different GDP per capita levels of China's provinces in 2035 and 2050, as shown in Eq. 10 and Table A.6.

$$VSL_{r,year,type} = VSL_{China,2015,type} \times \left(\frac{GDPper_{r,year}}{GDPper_{China,2015}} \right)^{elasticity} \quad (10)$$

Where, $VSL_{r,year,type}$ is the calibration medium/high/low VSL of China's provinces in 2035 and 2050, $VSL_{China,2015,type}$ is the medium/high/low VSL of China in 2015 estimated by Jin et al.(Jin et al., 2018), $GDPper_{China,2015}$ and $GDPper_{r,2050}$ are the per capita GDP of China in 2015 and the per capita GDP of China's provinces in 2050 under SSP2(Mondal et al., 2021), respectively. $elasticity$ is the elasticity coefficient of VSL on per capita GDP, and we adopt the reference value of 0.8(OECD, 2012). ”

Table A.6 Provincial VSL in 2035 and 2050 at the medium/high/low level (million US\$, 2020 price level)

Region	2035			2050		
	High VSL	Medium VSL	Low VSL	High VSL	Medium VSL	Low VSL
Anhui	1.59	0.95	0.70	2.23	1.34	0.98
Beijing	3.41	2.05	1.50	4.30	2.58	1.89
Chongqing	2.21	1.33	0.97	2.71	1.63	1.19
Fujian	3.38	2.03	1.49	4.79	2.88	2.11
Gansu	1.30	0.78	0.57	1.96	1.18	0.86
Guangdong	3.15	1.89	1.39	3.93	2.36	1.73
Guangxi	1.13	0.68	0.50	1.28	0.77	0.56
Guizhou	0.96	0.57	0.42	1.02	0.61	0.45
Hainan	1.92	1.15	0.84	2.34	1.40	1.03
Hebei	2.33	1.40	1.03	3.03	1.82	1.33
Heilongjiang	1.59	0.96	0.70	2.08	1.25	0.92
Henan	2.50	1.50	1.10	3.50	2.10	1.54
Hubei	2.39	1.43	1.05	3.17	1.90	1.40
Hunan	1.83	1.10	0.80	2.54	1.52	1.12
Inner Mongolia	3.13	1.88	1.38	3.67	2.20	1.62
Jiangsu	4.82	2.89	2.12	5.68	3.41	2.50
Jiangxi	1.72	1.03	0.76	2.05	1.23	0.90
Jilin	2.12	1.27	0.93	3.16	1.90	1.39
Liaoning	2.75	1.65	1.21	4.05	2.43	1.78
Ningxia	1.70	1.02	0.75	2.31	1.39	1.02
Qinghai	1.61	0.96	0.71	2.13	1.28	0.94
Shaanxi	1.77	1.06	0.78	2.71	1.62	1.19
Shandong	4.17	2.50	1.83	5.95	3.57	2.62
Shanghai	7.07	4.24	3.11	9.94	5.96	4.37
Shanxi	1.43	0.86	0.63	1.94	1.16	0.85
Sichuan	2.19	1.32	0.96	2.68	1.61	1.18

Tianjin	4.03	2.42	1.77	5.40	3.24	2.38
Xinjiang	1.40	0.84	0.62	1.84	1.10	0.81
Tibet	1.25	0.75	0.55	1.49	0.90	0.66
Yunnan	1.40	0.84	0.62	1.61	0.97	0.71
Zhejiang	4.72	2.83	2.08	6.50	3.90	2.86

Similar to recent studies (Zhou et al., 2021;Zhang and Chen, 2022), the transformation costs refer to the additional annualized capital costs and energy costs of various cooking and heating options in the rural residential sector in CNS compared to BaU. Capital costs are what households must pay to purchase the devices. Furthermore, we calculate the annualized cost based on the expected lifespan of the devices and the private discount rate (15%) for the residential sector (Jeuland and Pattanayak, 2012;Zhang et al., 2017). Annualized energy costs are what households must pay for cooking and heating fuels (including electricity, natural gas, LPG, clean coal and coal) each year. By comparing the system costs in different scenarios, we calculate the annual transformation costs of RCH achieving carbon neutrality (see following Eq. 3 and Eq. 4).

$$AnTransCost_{r,y} = AnSystemCost_{r,y,CNS} - AnSystemCost_{r,y,BaU} \quad (3)$$

$$AnSystemCost_{r,y,scn} = AnCapitalCost_{r,y,scn} + AnEnergyCost_{r,y,scn} \quad (4)$$

Where, $AnTransCost_{r,year}$ represents the annual transformation costs of RCH achieving carbon neutrality of province r in y year; $AnSystemCost_{r,y,scn}$ represents the annual energy system costs of RCH for province r in y year under the BaU or CNS scenario; $AnCapitalCost_{r,y,scn}$ and $AnEnergyCost_{r,y,scn}$ represent the annual capital and energy costs of RCH for province r in y year under the BaU or CNS scenario, respectively.

Based on your comments, we have added this detailed information in Manuscript Line 446.

“In our study, the key outputs of IMED|TEC include energy use, technology mix, emissions and associated system costs of RCH at the provincial level (Fig. 6). By comparing the system costs in different scenarios, we calculate the annualized transformation costs of RCH achieving carbon neutrality, including annualized capital costs and energy costs of various cooking and heating options in the rural residential sector. Capital costs are what households must pay to purchase the devices. Furthermore, the annualized capital cost is calculated based on the expected lifespan of the devices and the above private discount rate for the residential sector. Annualized energy costs are what households must pay for commercial fuels for cooking and heating each year.”

Reference

1. Jin Y, Zhang S. An Economic Evaluation of the Health Effects of Reducing Fine Particulate Pollution in Chinese Cities. *Asian Development Review* 35, 58-84 (2018).
2. Yin H, et al. Population ageing and deaths attributable to ambient PM2.5 a of economic cost. *Lancet Planetary Health* 5, E356-E367 (2021).
3. Gao AF, et al. Health and economic losses attributable to PM2.5 and ozone exposure in Handan, China. *Air Quality Atmosphere and Health* 14, 605-615 (2021).
4. Yan ML, et al. The exceptional heatwaves of 2017 and all-cause mortality: An assessment of nationwide health and economic impacts in China. *Science of the Total Environment* 812, (2022).
5. Mondal SK, et al. Doubling of the population exposed to drought over South Asia: CMIP6

- multi-model-based analysis. *The Science of the Total Environment* 771, 145186 (2021).
6. OECD. Mortality Risk Valuation in Environment, Health and Transport Policies (2012).
 7. Zhou M, et al. Environmental benefits and household costs of clean heating options in northern China. *Nature Sustainability* 5, 329-338 (2021).
 8. Zhang S, Chen WY. Assessing the energy transition in China towards carbon neutrality with a probabilistic framework. *Nature Communications* 13, (2022).
 9. Zhang W, Stern D, Liu X, Cai W, Wang C. An analysis of the costs of energy saving and CO₂ mitigation in rural households in China. *Journal of Cleaner Production* 165, 734-745 (2017).
 10. Jeuland MA, Pattanayak SK. Benefits and Costs of Improved Cookstoves: Assessing the Implications of Variability in Health, Forest and Climate Impacts. *Plos One* 7, (2012).

10. As a forward-looking simulation, the authors are also recommended to conduct a sensitivity analysis in addition to the central scenario in the main text.

Response:

Based on the one-at-a-time and two-at-a-time methods, we have conducted a systematic sensitivity analysis (116 sensitivity scenarios) with different assumptions on rural socioeconomic, residential energy systems and health impact evaluation. We covered alternative assumptions on key parameters related to the population, urbanization, GDP, modern technology efficiency and capital cost, modern energy cost, exposure-response functions (ERFs) and value of statistical life (VSL), to assess the robustness of our results. The sensitivity analysis was performed covering six groups of key input variables, including (1) different shared socioeconomic pathways, covering population, urbanization and GDP, related to different rural cooking and heating demand at the provincial level; (2) higher or lower modern technology capital cost; (3) higher or lower modern technology using efficiency; (4) higher or lower modern energy price; (5) different exposure-response functions; (6) higher or lower VSL. These parameters significantly affect rural residential costs, energy use, emissions, attributable PM_{2.5} concentrations and related health impacts.

Sensitivity analysis shows China's rural cooking and heating energy use, CO₂ and SO₂ emissions, and energy system costs are relatively sensitive to rural socioeconomic development. In addition, total energy use, electricity use and SO₂ emissions are moderately sensitive to the efficiency of the electric cooking range, while energy system cost is relatively sensitive to the capital cost of AAHP. Nonetheless, the model results still maintain significant differences between the corresponding sensitivity scenarios in BaU and CNS, i.e., China's RCH consistently would use more modern energy and emits less CO₂ and air pollutants while paying higher system costs for achieving carbon neutrality in 2060.

Sensitivity analysis results under the combinations of rural socioeconomic development, technology efficiency, modern technology capital cost, and modern energy price would lead to more sensitive results compared with other scenarios. However, even in the most sustainable combination (CNS_SSP3_EFoE&G[L]_ICoE&G[H]_EPoE&G[H], switching population and economic development pathway from SSP2 to SSP1, increasing the efficiency and decreasing the capital cost of electric cooking range and AAHP, decreasing modern energy price) or most unsustainable combination (CNS_SSP1_EFoE&G[H]_ICoE&G[L]_EPoE&G[L], switching population and economic development pathway from SSP2 to SSP3 and other settings are the opposite of the most sustainable combination) of carbon neutral scenarios, we found that the provincial PM_{2.5} reductions

differ little compared with the core CNS scenario.

In summary, the sensitivity results show that even though the alternative quantitative impacts differ from BaU and CNS, our conclusion that the benefits of the rural modern energy transition to carbon neutrality outpace the costs significantly is plausible.

More information about sensitivity analysis and the full suite of input parameters used are described in Supplementary Information - Appendix C, all sensitivity scenarios and detailed results are shown in Supplementary Sheet.

11. Please unify the text fonts in images, such as Figure 5 and others.

Response:

Thanks. We have unified the text fonts in all images in the Manuscript and Supplementary Information.

Fig. 5. Rural cooking and heating transformation costs and monetized health benefits related to avoided PM_{2.5}-associated premature deaths at the medium VSL level in 2035 and 2050.

12. Please increase the font size in Figure 3a-3d to match the font size of the other figures.

Response:

The font size in Figure 3a-3d has been increased to match the font size of the other figures.

Fig. 3. Changes in RCH emissions in rural China during 2014-2060 for (a) CO₂, (b) SO₂, (c) NO_x and (d) PM_{2.5} emissions, ambient PM_{2.5} concentrations reduction, avoided premature deaths in CNS relative to BaU (e) and regional disparity in carbon reduction, air quality improvement, and health benefits (f) in 2050.

13. Figure 6 needs improvement, as its meaning is not very clear.

Response:

We redrew the framework figure (Figure 6) to illustrate the three models used (IMED|TEC, GAINS and IMED|HEL) and their "soft-link", and highlighted the key inputs and outputs of each model.

Fig. 6. Integrated modeling assessment framework.

Reviewer #2 (Remarks to the Author):

Summary of review: The paper captures a very important but often under studied policy issue: how rural population achieve clean energy transition to achieve carbon neutrality and what are the costs around it. The energy-air pollution-human health modeling framework is comprehensive, and the results are insightful and policy relevant.

Response:

Thank you for your positive comments and suggestions, as they have been greatly helpful in enhancing our article.

A few questions and comments for the paper to consider:

1. The analysis relies on a few key assumptions: rural population, rural household energy demand, and future energy prices. I would suggest the authors move some of the key assumptions into the main text, and state the rationale for those assumptions clearly. Other assumptions could be included in the SI.

Response:

Our study includes several vital assumptions in our model, such as rural population and rural cooking and heating demand. **We have moved these key assumptions into the main text to make our manuscript more transparent, as shown in Method Table.1 in Manuscript. Additionally, we added other key assumptions, such as technology efficiency, energy price, emission factors and VSL in Supplementary Information - Appendix A.**

Table.1 in Manuscript – Method:

Table.1 China's projected rural population, cooking and heating service demand under SSP2

	2014	2020	2030	2040	2050	2060
Rural population (Billion people)	0.61	0.57	0.48	0.41	0.36	0.32
Cooking demand (Mtoe)	24	23	19	17	14	13
Heating demand (Mtoe)	22	23	22	20	18	17

Table A.1-A.4 and Table A.6 in Supplementary Information - Appendix A:

Table A.1 Technologies for rural cooking and heating in IMED|TEC and parameters

Number	Technology	Energy	Service	Efficiency
1	Straw cooking range	Straw	Cooking	0.21
2	Firewood cooking range	Firewood	Cooking	0.24
3	Coal cooking range	Coal	Cooking	0.28
4	Biogas cooking range	Biogas	Cooking	0.45
5	LPG cooking range	LPG	Cooking	0.55
6	Electric cooking range	Electricity	Cooking	0.85
7	Straw stove	Straw	Heating	0.24
8	Firewood stove	Firewood	Heating	0.28
9	Coal stove	Coal	Heating	0.39
10	Improved coal stove	Clean coal	Heating	0.71
11	Biogas stove	Biogas	Heating	0.57
12	Natural gas stove	Natural gas	Heating	0.85

13	Resistance heater	Electricity	Heating	0.90 0.07×ambient temperature (°C) +2.69
14	Air-to-air heat pump	Electricity	Heating	

Table A.2. Provincial efficiency of AAHP and ambient temperatures

Provinces/ municipalities/ autonomous regions	Average ambient temperatures during the heating season*	Efficiency of AAHP
Anhui	6.41	3.14
Beijing	0.4	2.72
Chongqing	10	3.39
Fujian	13.94	3.67
Gansu	-0.61	2.65
Guangdong	16.12	3.82
Guangxi	14.98	3.74
Guizhou	6.93	3.18
Hainan	20	4.09
Hebei	1.9	2.82
Heilongjiang	-12.33	1.83
Henan	4.37	3.00
Hubei	7.22	3.2
Hunan	8.55	3.29
Inner Mongolia	-6.47	2.24
Jiangsu	6.48	3.14
Jiangxi	9.18	3.33
Jilin	-9.86	2.00
Liaoning	-6.83	2.21
Ningxia	-2.84	2.49
Qinghai	-4.79	2.35
Shaanxi	3.51	2.94
Shandong	3.01	2.9
Shanghai	8.25	3.27
Shanxi	-1.23	2.6
Sichuan	8.78	3.3
Tianjin	0.41	2.72
Tibet	2.08	2.84
Xinjiang	-7.83	2.14
Yunnan	10.88	3.45
Zhejiang	8.22	3.27

*Average ambient temperatures during the heating seasons from 2010-2017 (November, 2010-February, 2018) come from the national meteorological observatory database. This database is

managed by China Meteorological Data Service Center.

Table A.3 Commercial energy prices without subsidies for the residential sector during 2014-2060 (US\$/toe, 2020 price level))

Fuel	2014	2020	2030	2040	2050	2060
coal	140	146	171	206	257	320
clean coal	280	291	342	412	512	638
NG/ LPG	370	385	429	474	540	611
electricity	770	755	701	639	570	508

Table A.4 Emission factors

Fuel	CO ₂ (kg/kgoe)	SO ₂ (g/kgoe)	NO _x (g/kgoe)	PM _{2.5} (g/kgoe)
Coal	3.96	24.16	3.22	16.10
Clean coal	3.47	3.38	1.90	0.14
Firewood	0.00	0.25	3.50	19.25
Straw	0.00	0.83	5.54	18.56
Biogas	0.14	1.68	0.48	0.24
Natural gas	2.35	0.42	0.84	0.00
LPG	2.64	0.17	1.75	0.42

Table A.6 Provincial VSL in 2035 and 2050 at the medium/high/low level (million US\$, 2020 price level)

Region	2035			2050		
	High VSL	Medium VSL	Low VSL	High VSL	Medium VSL	Low VSL
Anhui	1.59	0.95	0.70	2.23	1.34	0.98
Beijing	3.41	2.05	1.50	4.30	2.58	1.89
Chongqing	2.21	1.33	0.97	2.71	1.63	1.19
Fujian	3.38	2.03	1.49	4.79	2.88	2.11
Gansu	1.30	0.78	0.57	1.96	1.18	0.86
Guangdong	3.15	1.89	1.39	3.93	2.36	1.73
Guangxi	1.13	0.68	0.50	1.28	0.77	0.56
Guizhou	0.96	0.57	0.42	1.02	0.61	0.45
Hainan	1.92	1.15	0.84	2.34	1.40	1.03
Hebei	2.33	1.40	1.03	3.03	1.82	1.33
Heilongjiang	1.59	0.96	0.70	2.08	1.25	0.92
Henan	2.50	1.50	1.10	3.50	2.10	1.54
Hubei	2.39	1.43	1.05	3.17	1.90	1.40
Hunan	1.83	1.10	0.80	2.54	1.52	1.12
Inner Mongolia	3.13	1.88	1.38	3.67	2.20	1.62
Jiangsu	4.82	2.89	2.12	5.68	3.41	2.50
Jiangxi	1.72	1.03	0.76	2.05	1.23	0.90

Jilin	2.12	1.27	0.93	3.16	1.90	1.39
Liaoning	2.75	1.65	1.21	4.05	2.43	1.78
Ningxia	1.70	1.02	0.75	2.31	1.39	1.02
Qinghai	1.61	0.96	0.71	2.13	1.28	0.94
Shaanxi	1.77	1.06	0.78	2.71	1.62	1.19
Shandong	4.17	2.50	1.83	5.95	3.57	2.62
Shanghai	7.07	4.24	3.11	9.94	5.96	4.37
Shanxi	1.43	0.86	0.63	1.94	1.16	0.85
Sichuan	2.19	1.32	0.96	2.68	1.61	1.18
Tianjin	4.03	2.42	1.77	5.40	3.24	2.38
Xinjiang	1.40	0.84	0.62	1.84	1.10	0.81
Tibet	1.25	0.75	0.55	1.49	0.90	0.66
Yunnan	1.40	0.84	0.62	1.61	0.97	0.71
Zhejiang	4.72	2.83	2.08	6.50	3.90	2.86

2. The paper designed two scenarios: BaU and CNS. CNS makes sense as it is a stated policy scenario. However, BaU assumes "extrapolated from historical trends (1995-2014)" which probably needs better justification.

Response:

We have justified the BaU setting of our model as (Line 496):

"BaU is the reference scenario without carbon emissions constraints, in which rural cooking and heating would undergo a slow technology switching from 2015 to 2060 extrapolated from historical trends (1995-2014). We consider that the technology transformation process in BaU is mainly driven spontaneously by socioeconomic development, following energy ladder theory (Hosier and Dowd, 1987) and energy stacking theory (Masera et al., 2000)."

Therefore, BaU is a reference scenario used for comparison with the carbon neutrality scenario, which considers the spontaneous transition trends in rural cooking and heating at the provincial level. The energy ladder theory (Hosier et al., 1987) and energy stacking theory (Masera et al., 2000) state that as socioeconomic status increases, household energy would transform from primitive fuels (e.g., firewood and straw) to transitional fuels (e.g., coal) and further to modern fuels (e.g., electricity and LPG) in developing countries (Han et al., 2018). According to a retrospective survey of China's nationwide rural residential energy consumption, technology structure (Zhu et al., 2019) and energy mix (Tao et al., 2018) are undergoing an autonomous transformation towards efficiency and cleanliness, driven by the rising living standards of rural residents.

In contrast to CNS, BaU assumes an autonomous but slow rural cooking and heating technology and energy switching ("extrapolated from historical trends (1995-2014)") from traditional solid fuels to modern energy, which is a dynamic reference scenario. Additionally, our BaU is closer to reality than other studies (Ma et al., 2021; Xing et al., 2021), with fixed technology structure at the base year level in the baseline scenario.

Reference

1. Hosier RH, Dowd J. Household fuel choice in Zimbabwe-an empirical-test of the energy ladder hypothesis. *Resources and Energy* 9, 347-361 (1987).
2. Masera OR, Saatkamp BD, Kammen DM. From linear fuel switching to multiple cooking strategies: A critique and alternative to the energy ladder model. *World Development* 28, 2083-2103 (2000).
3. Han H, et al. Factors underlying rural household energy transition: A case study of China. *Energy Policy* 114, 234-244 (2018).
4. Zhu X, et al. Stacked Use and Transition Trends of Rural Household Energy in Mainland China. *Environmental Science & Technology* 53, 521-529 (2019).
5. Tao S, et al. Quantifying the rural residential energy transition in China from 1992 to 2012 through a representative national survey. *Nature Energy* 3, 567-573 (2018).
6. Xing R, et al. Deep decarbonization pathways in the building sector: China's NDC and the Paris agreement. *Environmental Research Letters* 16, (2021).
7. Ma S, et al. Roadmap towards clean and low carbon heating to 2035: A provincial analysis in northern China. *Energy* 225, 120164 (2021).

3. Does the paper consider "stock turnover" of the cooking and heating technologies? Not sure where the technology structural changes come from. Are they part of modeling results? Or are they assumptions the authors construct? If they are the latter, is there any research to support those assumptions?

Response:

We consider the "stock turnover" of cooking and heating technologies and technology structural changes are part of our modeling results.

In our study, there is a dynamic balance of technology stock quantity ("stock turnover") in the residential module of IMED|TEC. The stock quantity of cooking and heating technology t is calculated by the following Eq. 5. (See the Supplementary Information – Appendix A.1.2)

“(5) Dynamic balance of technology stock quantity

The stock quantity of technology t is calculated by Eq. 5.

$$S_t = SS_t \cdot \left(1 - \frac{1}{T_t}\right) - w_t + r_t \quad (5)$$

Where, S_t represents the operating quantity of technology t in the current year and SS_t represents the remaining stock quantity technology t in the previous year, respectively; w_t represents the retired stock quantity of technology t before lifetime; r_t represents recruit quantity of technology t to meet the service demand. S_t and SS_t follow the Weibull distribution. $SS_t \left(1 - \frac{1}{T_t}\right)$ represents the remaining stock quantity from the previous year to the current year after natural depreciation.”

For instance, when the carbon emission cap forces rural heating to transition from coal to natural gas, coal stoves would not only naturally depreciate following the Weibull distribution, part of them would retire before their lifetime. Correspondingly, a certain amount of natural gas stoves would be recruited to meet the heating demand gap. This is an intuitive technology structural changes from coal stoves to natural gas stoves.

4. I do not see any in-depth discussions on how the results are sensitive to some key assumptions, such as urbanization, fuel structure, fuel prices, VSL, etc.

Response:

Based on the one-at-a-time and two-at-a-time methods, we have conducted a systematic sensitivity analysis (116 sensitivity scenarios) with different assumptions on rural socioeconomic, residential energy systems and health impact evaluation. We covered alternative assumptions on key parameters related to the population, urbanization, GDP, modern technology efficiency and capital cost, modern energy cost, exposure-response functions (ERFs) and value of statistical life (VSL), to assess the robustness of our results. The sensitivity analysis was performed covering six groups of key input variables, including (1) different shared socioeconomic pathways, covering population, urbanization and GDP, related to different rural cooking and heating demand at the provincial level; (2) higher or lower modern technology capital cost; (3) higher or lower modern technology using efficiency; (4) higher or lower modern energy price; (5) different exposure-response functions; (6) higher or lower VSL. These parameters significantly affect rural residential costs, energy use, emissions, attributable PM_{2.5} concentrations and related health impacts. Notably, fuel structure is part of modeling results, not an input assumption, so we did not conduct a sensitivity analysis about it.

Sensitivity analysis shows China's rural cooking and heating energy use, CO₂ and SO₂ emissions, and energy system costs are relatively sensitive to rural socioeconomic development. In addition, total energy use, electricity use and SO₂ emissions are moderately sensitive to the efficiency of the electric cooking range, while energy system cost is relatively sensitive to the capital cost of AAHP. Nonetheless, the model results still maintain significant differences between the corresponding sensitivity scenarios in BaU and CNS, i.e., China's RCH consistently would use more modern energy and emits less CO₂ and air pollutants while paying higher system costs for achieving carbon neutrality in 2060.

Sensitivity analysis results under the combinations of rural socioeconomic development, technology efficiency, modern technology capital cost and modern energy price would lead to more sensitive results compared with other scenarios. However, even in the most sustainable combination (CNS_SSP3_EFoE&G[L]_ICoE&G[H]_EPoE&G[H]), when switching population and economic development pathway from SSP2 to SSP1, increasing the efficiency and decreasing the capital cost of electric cooking range and AAHP, decreasing modern energy price) or most unsustainable combination (CNS_SSP1_EFoE&G[H]_ICoE&G[L]_EPoE&G[L], switching population and economic development pathway from SSP2 to SSP3 and other settings are the opposite of the most sustainable combination) of carbon neutral scenarios, we found that the provincial PM_{2.5} reductions differ little compared with the core CNS scenario.

In summary, the sensitivity results show that even though the alternative quantitative impacts differ from BaU and CNS, our conclusion that the benefits of the rural modern energy transition to carbon neutrality outpace the costs significantly is plausible.

More information about sensitivity analysis and the full suite of input parameters used are described in Supplementary Information - Appendix C, all sensitivity scenarios and detailed results are shown in Supplementary Sheet.

5. Other minor suggestions:

a. Ln41, better to give a rural population number in the same year.

Response:

Based on your comments, we have added the number of the rural population in Line 41 as:
"China is home to the second largest rural population worldwide, with approximately 0.55 billion individuals consuming about 163 million tons of oil equivalent (Mtoe) of commercial energy in the residential sector in 2019 (NBSC, 2020;Wang et al., 2022)."

Reference

1. Wang N, et al. Spatial-temporal variation and coupling analysis of residential energy consumption and economic growth in China. *Applied Energy* **309**, (2022).
2. NBSC. *China Statistical Yearbook 2019*. (2020).

b. Try to avoid using "Without a doubt (Ln49)".

Response:

We have replaced "Without a doubt" with "Therefore". We have also checked the manuscript and ensured it avoids using the same expression.

c. The main text needs to better cite appendix tables and assumptions.

Response:

Thanks. We have correctly and properly cited the tables and assumptions in Supplementary Information.

Reviewer #3 (Remarks to the Author):

This is a very interesting and meaningful topic, focusing on the energy transition in the rural area, with useful discussions on energy poverty, inequality, health impact. I have some comments as below.

Response:

Thank you for your overall encouragement and valuable comments below, they are very insightful and helpful to improve our work.

1. In the Introduction section, are you suggesting that China's rural residential energy use is accounting for "4.7% of China's total energy consumption" but carbon emissions from the rural residential sector is accounting for "42.1% of China's total residential energy-related CO2 emissions"? Why not show the proportion of "energy use in China's rural residential sector" in "energy use in China residential sector"? So that would help us understand the difference in carbon intensity between China's rural area and China as a whole.

Response:

Following your suggestion, in order to make the quantitative statements more straightforward, we have revised the original contents as Line 47 in the Manuscript:

"energy use for cooking and heating in China's rural residential sector accounts for 72% of China's total residential cooking and heating energy use (Yun et al., 2020)."

Our study focuses on heating and cooking in the rural residential sector because energy consumption in rural residential is primarily attributed to cooking and heating, which account for over 80% of rural residential energy consumption (Zheng, 2015;Wu et al., 2017). Therefore, in response to your comments, we have compared the proportion of energy used for cooking and heating by rural residents with that of the residential sector in China. We found that energy use for cooking and heating in China's rural residential sector accounts for 72% of China's total residential cooking and heating energy use (Yun et al., 2020).

Reference

1. Yun X, et al. Residential solid fuel emissions contribute significantly to air pollution and associated health impacts in China. *Science Advances* 6, (2020).
2. Wu S, et al.. Measurement of inequality using household energy consumption data in rural China. *Nature Energy* 2, 795-803 (2017).
3. Zheng X. China Household Energy Consumption Research Report. Science Press (2015).

2. In the Introduction section, why "switching rural residential energy toward modern energy is pivotal to simultaneously attain multi-Sustainable Development Goals (SDGs) of rural revitalization (SDG 1, SDG 10)"? it doesn't seem very straight forward to me, and you also say the energy transition is "economic burdens" in page

Response:

We have clearly articulated the relationship between modern energy usage and SDGs in the Introduction section, as shown in Line 50 in the Manuscript:

"Therefore, switching rural residential energy toward modern energy is pivotal to simultaneously attain multi Sustainable Development Goals (SDGs) of clean energy (SDG 7), rural

revitalization driven by energy poverty improvement (SDG 1) and universal modern energy access (SDG 10), carbon neutrality (SDG 13), better air quality related good health (SDG 3).”

Based on the energy development index (EDI) developed by the International Energy Agency, modern energy share is one key indicator to characterize energy poverty (IEA, 2010). Therefore, residential energy transitions from solid fuels toward modern energy could increase the modern energy share of households and improve energy poverty (related to SDG 1). Additionally, carbon neutrality targets have pushed each province to transition to modern energy for rural cooking and heating, resulting in universal modern energy access. This would help to improve the current unequal patterns (Li et al., 2019) of developed regions using more modern energy (related SDG 10).

The "economic burdens" in our manuscript just mean that residents need to pay more money for modern equipment and energy. As shown in Line 287 in the Manuscript “the unsubsidized capital cost of AAHP is ~3000 US\$ per unit (Zhou et al., 2021)”, rural residents may spend more on advanced equipment to reach carbon neutrality, which will bring economic burdens on rural residents. This perspective is consistent with previous studies (Zhang et al., 2022). However, it should be noted that expenditure or investment in rural technological advancement should not be only regarded as an economic burden. Instead, it is a key indicator of living standard improvement that narrows the gap and reduces inequality compared with urban areas and could be regarded as rural revitalization. Additionally, the investment need for modern cooking and heating technologies could also drive economic growth.

Reference

1. IEA. *World Energy Outlook 2010* (2010).
2. Li JL, et al. Transition from non-commercial to commercial energy in rural China: Insights from the accessibility and affordability. *Energy Policy* 127, 392-403 (2019).
3. Zhou M, et al. Environmental benefits and household costs of clean heating options in northern China. *Nature Sustainability* 5, 329-338 (2021).
4. Zhang S, Chen WY. Assessing the energy transition in China towards carbon neutrality with a probabilistic framework. *Nature Communications* 13, (2022).

3. In the Introduction section, data is a bit out of date. Many pollution related data is date back to 2012, more than 10 years ago. Would be great to provide an update.

Response:

We have updated on rural residential energy use attributable to pollution as Line 48 in the Manuscript:

"Meanwhile, rural residential energy use contributed to disproportionately high levels of air pollutants emissions, leading to over 3.9 $\mu\text{g}/\text{m}^3$ ambient $\text{PM}_{2.5}$ in 2014 (Yun et al., 2020) that harms human health(Zhang and Smith, 2007;Duan et al., 2014;Shen et al., 2019;Gu et al., 2020;Yun et al., 2020)."

As we also responded to Reviewer #1 comments #1 and 2, constrained by the considerable workload and cost of conducting a nationwide survey for rural household energy use, there are no more recent national rural household energy surveys that can better represent the provincial level rural household energy consumption in China than we used in our study. This representative nationwide rural household energy database was conducted by Tao et al. and the most recent historical data is updated to 2014 (Tao et al., 2018). This dataset was widely used in recent high-

impact studies (Tao et al., 2018;Shen et al., 2019;Zhu et al., 2019;Yun et al., 2020;Meng et al., 2021) and is considered to be the most precise nationwide rural household energy data in China. For instance, Yun et al.(Yun et al., 2020) found that rural cooking and heating contributed to the national rural and urban ambient PM_{2.5} concentration in 2014 was 3.9 and 4.9 µg/m³, respectively. In addition, recent studies based on other emission inventories also estimated nationwide rural residential attributable air pollution around 2014 (Zhao et al., 2019;Lu et al., 2022).

Reference

1. Yun X, et al. Residential solid fuel emissions contribute significantly to air pollution and associated health impacts in China. *Science Advances* 6, (2020).
2. Shen G, et al. Impacts of air pollutants from rural Chinese households under the rapid residential energy transition. *Nature Communications* 10, 3405-3408 (2019).
3. Gu Y, Zhang W, Yang Y, Wang C, Streets DG, Yim SHL. Assessing outdoor air quality and public health impact attributable to residential black carbon emissions in rural China. *Resources Conservation and Recycling* 159, (2020).
4. Zhang JJ, Smith KR. Household air pollution from coal and biomass fuels in China: Measurements, health impacts, and interventions. *Environmental Health Perspectives* 115, 848-855 (2007).
5. Duan XL, et al. Household fuel use for cooking and heating in China: Results from the first Chinese Environmental Exposure-Related Human Activity Patterns Survey (CEERHAPS). *Applied Energy* 136, 692-703 (2014).
6. Tao S, et al. Quantifying the rural residential energy transition in China from 1992 to 2012 through a representative national survey. *Nature Energy* 3, 567-573 (2018).
7. Meng W, et al. Synergistic Health Benefits of Household Stove Upgrading and Energy Switching in Rural China. *Environmental Science & Technology* 55, 14567-14575 (2021).
8. Zhao H, et al. Inequality of household consumption and air pollution-related deaths in China. *Nature Communications* 10, 4337-4339 (2019).
9. Lu C, et al. Reduced health burden and economic benefits of cleaner fuel usage from household energy consumption across rural and urban China. *Environmental Research Letters* 17, 14039 (2022).

4. Methodology: for the residential module of IMED|TEC model, have you considered the reduction in costs brought about by technological progress? And in particular for electricity, it is more complicated as it will be determined by how the electricity is generated (i.e. the power generation mix). Has the model considered different technologies for electricity generation?

Response:

Our study considered the cost reduction brought about by technological progress. Technological progress (due to learning-by-doing, standardization and so on) results in cost reductions of modern technologies. In energy system simulation, "learning curves" generally inform future expectations for similar technologies (Zhang et al., 2022). Considering modern technologies in the residential sector (air-to-air heat pump, electrical cooking range, LPG cooking range and natural gas stove) have already gained significant learning experience, they

will follow a slower cost reduction trajectory. Therefore we set the costs of modern technologies to decline linearly by 20% between 2020 and 2060(IEA, 2020) in the residential module of IMED|TEC.

This research primarily focuses on the rural residential sector, and the related energy used for cooking and heating as direct energy like coal, biomass, biogas, electricity, etc. It demonstrated that the research boundary excluded energy production such as electricity generation. Therefore, this study did not consider different technologies for electricity generation. It would be an interesting direction to simultaneously represent power generation in future modeling improvement. However, we have considered the projected energy price trajectory. In the future, carbon emission limits lead to higher fossil energy prices and lower electricity prices, which is consistent with the current trend of technological development. Especially, we set the electricity price to decline by 33% between 2020 and 2060 (Cao et al., 2021;Liu et al., 2022) in the residential module of IMED|TEC.

Reference

1. Zhang S, Chen WY. Assessing the energy transition in China towards carbon neutrality with a probabilistic framework. *Nature Communications* 13, (2022).
2. IEA. Energy Technology Perspectives 2020. (2020).
3. Cao J, et al. The general equilibrium impacts of carbon tax policy in China: A multi-model comparison. *Energy Economics* 99, (2021).
4. Liu XY, et al. Achieving carbon neutrality enables China to attain its industrial water-use target. *One Earth* 5, 188-200 (2022).

Reference

- [1] Cao, J., Dai, H. C., Li, S. T., et al. 2021. The general equilibrium impacts of carbon tax policy in China: A multi-model comparison[J]. Energy Economics **99**.
- [2] Chen, Y., Guo, F., Wang, J., et al. 2020. Provincial and gridded population projection for China under shared socioeconomic pathways from 2010 to 2100[J]. Scientific Data **7**(1): 83.
- [3] Duan, X. L., Jiang, Y., Wang, B. B., et al. 2014. Household fuel use for cooking and heating in China: Results from the first Chinese Environmental Exposure-Related Human Activity Patterns Survey (CEERHAPS)[J]. Applied Energy **136**: 692-703.
- [4] EFC. Energy Foundation China, Beijing, China. . 2020. Synthesis Report 2020 on China's Carbon Neutrality: China's New Growth Pathway: from the 14th Five Year Plan to Carbon Neutrality. Available from: <https://www.efchina.org/Reports-en/report-lceg-20201210-en>. Accessed:
- [5] Gao, A. F., Wang, J. Y., Luo, J. F., et al. 2021. Health and economic losses attributable to PM_{2.5} and ozone exposure in Handan, China[J]. Air Quality Atmosphere and Health **14**(5): 605-615.
- [6] Gu, Y., Zhang, W., Yang, Y., et al. 2020. Assessing outdoor air quality and public health impact attributable to residential black carbon emissions in rural China[J]. Resources Conservation and Recycling **159**.
- [7] Han, H. Y., Wu, S. and Zhang, Z. J. 2018. Factors underlying rural household energy transition: A case study of China[J]. Energy Policy **114**: 234-244.
- [8] Hosier, R. H. and Dowd, J. 1987. Household fuel choice in Zimbabwe-an empirical-test of the energy ladder hypothesis[J]. Resources and Energy **9**(4): 347-361.
- [9] IEA 2020. Energy Technology Perspectives 2020[J].
- [10] Jeuland, M. A. and Pattanayak, S. K. 2012. Benefits and Costs of Improved Cookstoves: Assessing the Implications of Variability in Health, Forest and Climate Impacts[J]. Plos One **7**(2).
- [11] Jin, Y. and Zhang, S. 2018. An Economic Evaluation of the Health Effects of Reducing Fine Particulate Pollution in Chinese Cities[J]. Asian development review **35**(2): 58-84.
- [12] Li, J. L., Chen, C. and Liu, H. X. 2019. Transition from non-commercial to commercial energy in rural China: Insights from the accessibility and affordability[J]. Energy Policy **127**: 392-403.
- [13] Liu, X. Y., Dai, H. C., Wada, Y., et al. 2022. Achieving carbon neutrality enables China to attain its industrial water-use target[J]. One Earth **5**(2): 188-200.
- [14] Lu, C., Zhang, S., Tan, C., et al. 2022. Reduced health burden and economic benefits of cleaner fuel usage from household energy consumption across rural and urban China[J]. Environmental Research Letters **17**(1): 14039.
- [15] Ma, S., Guo, S., Zheng, D., et al. 2021. Roadmap towards clean and low carbon heating to 2035: A provincial analysis in northern China[J]. Energy **225**: 120164.
- [16] Masera, O. R., Saatkamp, B. D. and Kammen, D. M. 2000. From linear fuel switching to multiple cooking strategies: A critique and alternative to the energy ladder model[J]. World Development **28**(12): 2083-2103.
- [17] Meng, W. J., Shen, G. F., Shen, H. Z., et al. 2021. Synergistic Health Benefits of Household Stove Upgrading and Energy Switching in Rural China[J]. Environmental Science & Technology **55**(21): 14567-14575.
- [18] Mondal, S. K., Huang, J., Wang, Y., et al. 2021. Doubling of the population exposed to drought over South Asia: CMIP6 multi-model-based analysis[J]. The Science of the Total Environment **771**: 145186.
- [19] NBSC 2015. China Statistical Yearbook 2014[J].

- [20] NBSC 2020. China Statistical Yearbook 2019[J].
- [21] OECD (2012). Mortality Risk Valuation in Environment, Health and Transport Policies[M].
- [22] Shen, G., Ru, M., Du, W., et al. 2019. Impacts of air pollutants from rural Chinese households under the rapid residential energy transition[J]. Nature Communications **10**(1): 3405-3408.
- [23] Tao, S., Ru, M. Y., Du, W., et al. 2018. Quantifying the rural residential energy transition in China from 1992 to 2012 through a representative national survey[J]. Nature Energy **3**(7): 567-573.
- [24] Wang, N., Fu, X. D. and Wang, S. B. 2022. Spatial-temporal variation and coupling analysis of residential energy consumption and economic growth in China[J]. Applied Energy **309**.
- [25] Wu, S., Zheng, X. and Wei, C. 2017. Measurement of inequality using household energy consumption data in rural China[J]. Nature Energy **2**(10): 795-803.
- [26] Xing, R., Hanaoka, T. and Masui, T. 2021. Deep decarbonization pathways in the building sector: China's NDC and the Paris agreement[J]. Environmental Research Letters **16**(4).
- [27] Yan, M. L., Xie, Y., Zhu, H. H., et al. 2022. The exceptional heatwaves of 2017 and all-cause mortality: An assessment of nationwide health and economic impacts in China[J]. Science of the Total Environment **812**.
- [28] Yang, X. 2018. Current situations and technical routes of rural clean heating (in Chinese).[J]. The 14th session of Building Energy Efficiency Academic Week in Tsinghua University: Clean Heating Forum, Beijing.
- [29] Yin, H., Brauer, M., Zhang, J. F., et al. 2021. Population ageing and deaths attributable to ambient PM_{2.5} a of economic cost[J]. Lancet Planetary Health **5**(6): E356-E367.
- [30] Yun, X., Shen, G., Shen, H., et al. 2020. Residential solid fuel emissions contribute significantly to air pollution and associated health impacts in China[J]. Science Advances **6**(44).
- [31] Zhang, J. J. and Smith, K. R. 2007. Household air pollution from coal and biomass fuels in China: Measurements, health impacts, and interventions[J]. Environmental Health Perspectives **115**(6): 848-855.
- [32] Zhang, S. and Chen, W. Y. 2022. Assessing the energy transition in China towards carbon neutrality with a probabilistic framework[J]. Nature Communications **13**(1).
- [33] Zhang, W., Stern, D., Liu, X., et al. 2017. An analysis of the costs of energy saving and CO₂ mitigation in rural households in China[J]. Journal of Cleaner Production **165**: 734-745.
- [34] Zhao, H., Geng, G., Zhang, Q., et al. 2019. Inequality of household consumption and air pollution-related deaths in China[J]. Nature Communications **10**(1): 4337-4339.
- [35] Zheng, X. (2015). China Household Energy Consumption Research Report[M]. Beijing, Science Press.
- [36] Zhou, M., Liu, H., Peng, L., et al. 2021. Environmental benefits and household costs of clean heating options in northern China[J]. Nature Sustainability **5**(4): 329-338.
- [37] Zhu, X., Yun, X., Meng, W., et al. 2019. Stacked Use and Transition Trends of Rural Household Energy in Mainland China[J]. Environmental Science & Technology **53**(1): 521-529.

REVIEWERS' COMMENTS

Reviewer #1 (Remarks to the Author):

I appreciate the efforts by the authors to address my review comments. In particular, the systematic sensitivity analysis makes this study much more convincing. The paper is much improved and my comments are mostly well-addressed. Overall, it can be recommended for publishing provided that the following minor revisions are made:

1. Line 54: "Fortunately" >- "Historically"

Please avoid using subjective words in the manuscript.

2. Line 219: "northern China" >- "Northern China"

3. Line 292: the authors state that the additional transformation cost of decarbonization of rural cooking and heating is "equivalent to 41 US\$ per capita and 0.01% of GDP". Please provide the related references to explain how the per capita value and percentage value are estimated.

Reviewer #2 (Remarks to the Author):

The authors have carefully addressed my questions. I do not have further comments.

Dear Reviewers,

Many thanks for reviewing our manuscript. We have made revisions point by point after carefully reviewing the comments. Please find our response to the individual comments (shown in blue text). When showing changes to the text (Manuscript and Supplementary Information), new sentences/words are shown in purple in this response letter.

Reviewer #1 (Remarks to the Author):

I appreciate the efforts by the authors to address my review comments. In particular, the systematic sensitivity analysis makes this study much more convincing. The paper is much improved and my comments are mostly well-addressed. Overall, it can be recommended for publishing provided that the following minor revisions are made:

Response: Thank you for your positive evaluations, and the remaining comments have been addressed below.

1. Line 54: “Fortunately” >- “Historically”

Please avoid using subjective words in the manuscript.

Response:

We have replaced "Fortunately" with "Historically" ,and we have also checked the manuscript and avoided using subjective words.

2. Line 219: “northern China” >- “Northern China”

Response:

We have replaced "northern China" with "Northern China" in the line mentioned.

3. Line 292: the authors state that the additional transformation cost of decarbonization of rural cooking and heating is “equivalent to 41 US\$ per capita and 0.01% of GDP”. Please provide the related references to explain how the per capita value and percentage value are estimated.

Response:

Compared with BaU, the total additional transformation costs of deep decarbonization for rural cooking and heating in CNS is 13 billion US\$ in 2060. Under the well-established SSP2 framework, China's projected rural people and GDP are 0.32 billion persons (Chen et al., 2020) and 102 trillion US\$ (Jiang et al., 2018), respectively, in 2060.

In detail, we divided rural residential transformation costs (13 billion US\$) by projected rural people (0.32 billion) to calculate the per capita transformation cost of 41 US\$. Similarly, we also divided rural residential transformation costs (13 billion US\$) by projected GDP (102 trillion US\$) to calculate the ratio of 0.01%.

Based on your comments, we have added the related references in Line 312 of the Manuscript: “Compared with BaU, the total additional transformation costs of deep decarbonization for rural cooking and heating in CNS is 13 billion US\$ in 2060 (including 2 billion US\$ for cooking and 11 billion US\$ for heating), equivalent to 41 US\$ per capita (Chen et al., 2020)

and 0.01% of GDP (Jiang et al., 2018).”

Reference

1. Chen Y, et al. Provincial and gridded population projection for China under shared socioeconomic pathways from 2010 to 2100. *Scientific Data* **7**, 83 (2020).
2. Jiang T, et al. Projection of national and provincial economy under the shared socioeconomic pathways in China. *Progressus Inquisitiones de Mutatione Climatis* **14**, 50-58 (2018)

Reviewer #2 (Remarks to the Author):

The authors have carefully addressed my questions. I do not have further comments.

Response: Thank you for the positive evaluations.